# Early Stopping Against Label Noise Without Validation Data

**Suqin Yuan**[1]   **Lei Feng**[2]*   **Tongliang Liu**[1]*
[1] Sydney AI Centre, School of Computer Science, The University of Sydney
[2] School of Computer Science and Engineering, Nanyang Technological University

## Abstract

Early stopping methods in deep learning face the challenge of balancing the volume of training and validation data, especially in the presence of label noise. Concretely, sparing more data for validation from training data would limit the performance of the learned model, yet insufficient validation data could result in a sub-optimal selection of the desired model. In this paper, we propose a novel early stopping method called *Label Wave*, which *does not require validation data* for selecting the desired model in the presence of label noise. It works by tracking the changes in the model's predictions on the training set during the training process, aiming to halt training before the model unduly fits mislabeled data. This method is empirically supported by our observation that minimum fluctuations in predictions typically occur at the training epoch before the model excessively fits mislabeled data. Through extensive experiments, we show both the effectiveness of the *Label Wave* method across various settings and its capability to enhance the performance of existing methods for learning with noisy labels.

## 1 Introduction

Deep Neural Networks (DNNs) are praised for their remarkable expressive power, which allows them to uncover intricate patterns in high-dimensional data (Montufar et al., 2014; LeCun et al., 2015) and can even fit data with random labels. However, this strength, often termed *Memorization* (Zhang et al., 2017), can be a double-edged sword, especially when encountering label noise. When label noise exists, the inherent capability of DNNs might cause the model to fit mislabeled examples from noisy datasets, which can deteriorate its generalization performance. Specifically, when DNNs are trained on noisy datasets containing both clean and mislabeled examples, it is often observed that the test error initially decreases and subsequently increases. To prevent DNNs from overconfidently learning from mislabeled examples, many existing methods for learning with noisy labels (Xia et al., 2019; Han et al., 2020; Song et al., 2022; Huang et al., 2023) explicitly or implicitly adopted the operation of halting training before the test error increases—a strategy termed "early stopping".

Early stopping relies on model selection, aiming to choose a model that aligns most closely with the true concept from a range of candidate models obtained during the training process (Mohri et al., 2018; Bai et al., 2021). To this end, leveraging hold-out validation data to pinpoint an appropriate early stopping point for model selection becomes a prevalent approach (Xu & Goodacre, 2018) in deep learning. However, this approach heavily relies on additional validation data that is usually derived by splitting the training set, thereby resulting in degraded performance due to insufficient training data. This issue becomes even more challenging in the presence of label noise. This is because, in this scenario, it is hard to learn a good model from unreliable training data and select the desired model using unreliable validation data (Chen et al., 2021).

In this paper, we aim to pinpoint an appropriate early stopping point to mitigate the negative effects of label noise *without relying on any hold-out validation data*. Our motivation stems from the observation that there exist qualitative differences between DNNs trained on clean examples and mislabeled examples (Arpit et al., 2017). By tracking these differences during the training process, we can pinpoint the moment when DNNs transition from primarily fitting clean examples to primarily fitting mislabeled examples, and thus determine an appropriate early stopping point. In what

---

*Corresponding authors.

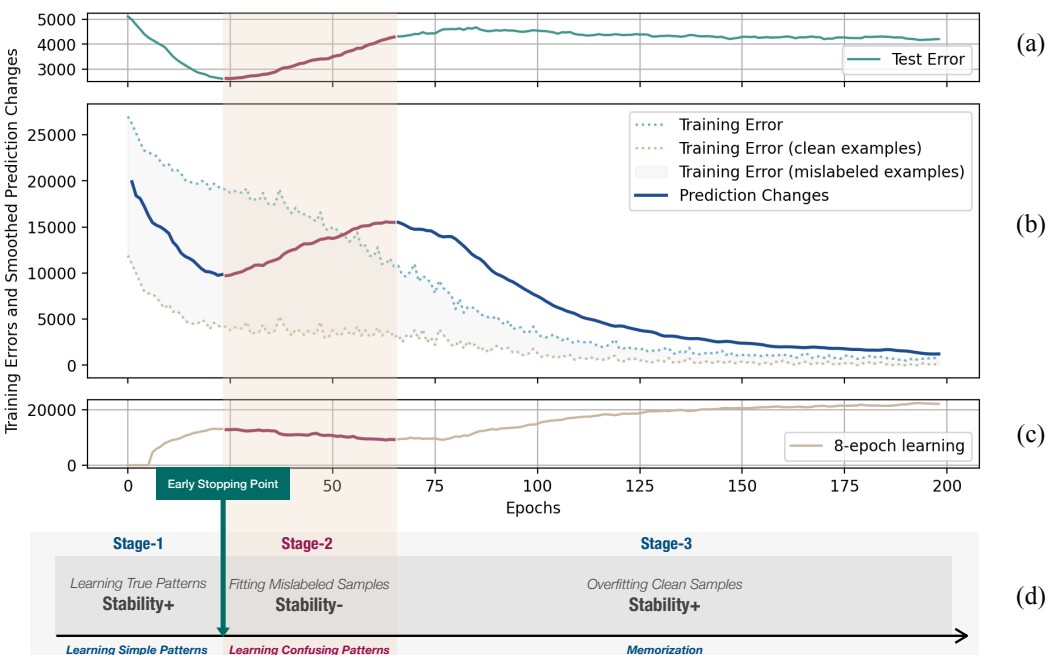

Figure 1: We examine how the model's fitting and generalization performance evolves during the training process of learning with noisy labels. Utilizing the *k-epoch learning* metric (Yuan et al., 2023), we measure the number of training examples that can be consistently classified according to their provided labels. This allows us to capture fluctuations in the model's fitting performance. We categorize the training process into three stages according to the stability of the fitting performance (panel d). This categorization is informed by an integrated analysis of generalization performance derived from *test error* (panel a) and fitting performance derived from *training error* (panel b) and 8-epoch learning metrics (panel c). Thus, we design the *prediction changes* metric to measure the shifts in the model's predictions on the training set to pinpoint the early stopping point. For detailed information regarding the experiment settings, please refer to Appendix B and Section 3.

follows, we demonstrate in Figure 1 how to determine such a point by tracking the changes in the model's *fitting performance*, instead of leveraging any validation with hold-out data.

To accurately pinpoint an appropriate early stopping point without relying on hold-out data, we introduce a metric called *prediction changes* (PC), as shown in Figure 1. We use PC to measure the number of examples whose predictions have changed between two successive epochs. In other words, this metric quantifies the prediction stability degree of the model trained on a noisy training set. Therefore, this metric offers a straightforward method for tracking the changes in the model's *fitting performance* throughout the training process. These changes, metaphorically likened to wind waves, inspired the name of our method, "*Label Wave*".

The effectiveness of the PC becomes evident when observing Stages 1 and 2 in Figure 1: the curve of PC in Figure 1(b) exactly matches the curve of *test error* in Figure 1(a). In Stage 1, as the model begins to fit clean examples (i.e., learns simple patterns), both the PC and the *test error* steadily decrease. In Stage 2, as depicted in *training error* in Figure 1(b), the model begins to fit mislabeled examples significantly, thereby leading to a rise in *test error*. Concurrently, the model's fitting performance experiences notable fluctuations or even declines. Hence there is an increase in the model's prediction fluctuations, leading to an increase in PC. Thus, the local minimum of PC, marking the turning point between Stages 1 and 2, is identified as the early stopping point.

In Stage 2, the rise in both PC and *test error* indicates that the model's efforts to minimize the total loss across the training set to accurately fit mislabeled examples not only undermine its ability to generalize to test data, but also harm its performance to fit the correct patterns in the training data. In other words, *fitting mislabeled examples impairs the overall model's fitting performance*. This pronounced phenomenon in Stage 2 is termed *learning confusion patterns*, reflecting the model's misinterpretation of the training data distribution.

Notably, our proposed *Label Wave* method can identify the early stopping point at which the model transitions from primarily fitting clean examples to mislabeled ones. This can be achieved without prior knowledge about which training examples are mislabeled examples or the need for separate hold-out validation data. This method not only provides a more efficient way to determine the early stopping point but also ensures sufficient training data to produce an effective model.

Our main contributions can be summarized as follows:

1. We present an empirical observation that, when training DNNs from a noisy dataset, fitting mislabeled examples impairs not only the generalization performance but also *the overall model's fitting performance*.

2. Building on the correlation between the model's generalization and fitting performance established in our empirical analysis, we propose the *Label Wave* method, which uses the changes in the model's predictions on the training set to identify the early stopping point.

3. Through extensive experiments, we show both the effectiveness of the *Label Wave* method across various settings and its capability to enhance the performance of existing methods for learning with noisy labels.

## 2 RELATED WORK

**Memorization & Forgetting.** In deep neural networks, generalization is not solely dictated by the complexity of the hypothesis space (Chaudhari et al., 2019; Advani et al., 2020; Jiang et al., 2020). They can still generalize effectively even without regularizers, a characteristic termed as *Memorization*[1]. Building on this, recent publications (Toneva et al., 2019; Feldman & Zhang, 2020) have studied the phenomenon known as the "*forgetting event*", aiming to understand how training data and network structures influence generalization. Based on the concept of *forgetting*, we observed that when the model starts to fit mislabeled examples, it begins to significantly forget the training data. This forgetting increases until it reaches a turning point, after which the forgetting decreases.

**Model Selection.** Numerous indirect methods have been proposed for selecting an appropriate model. These methods include marginal likelihood estimator (Duvenaud et al., 2016), computed gradients (Mahsereci et al., 2017), leveraging unlabeled data (Garg et al., 2021; Deng & Zheng, 2021; Forouzesh et al., 2023), noise stability (Arora et al., 2018; Morcos et al., 2018; Zhang et al., 2019), estimating generalization gap (Jiang et al., 2018; Corneanu et al., 2020), modeling loss distribution (Song et al., 2019b; 2021; Lu & He, 2022), and training speed (Lyle et al., 2020; Ru et al., 2021). In contrast to the existing methods, *Label Wave* focuses on the selection of an appropriate early stopping point during training process. Notably, this is achieved without the need for additional or hold-out data and requires no preprocessing, ensuring low computational overhead.

**Learning Stages.** Traditional paradigms often posit the presence of two stages during the learning process: underfitting and overfitting (Goodfellow et al., 2016; Lin et al., 2023). Researchers exploring models trained on randomly labeled examples have divided the learning process into two distinct stages for a more nuanced perspective on deep learning: an initial stage of "learning simple patterns" (Arpit et al., 2017), followed by a subsequent stage of "memorization" (Zhang et al., 2017). Further findings, such as epoch-wise Deep Double Descent (Belkin et al., 2019; Nakkiran et al., 2021) and "Grokking" (Power et al., 2022; Liu et al., 2022b; Nanda et al., 2023), have highlighted the potential existence of additional learning stages under specific conditions. However, these studies primarily focus on classifying learning stages based on generalization performance when models learn from imperfect data, thereby overlooking variations in model fitting performance.

**Learning Dynamics.** A wealth of research endeavors have aimed to classify the "hardness" of examples by tracking learning dynamics, which is particularly beneficial for scenarios involving noisy labels or long-tail distributions. Methods like Late Stopping (Yuan et al., 2023), FSLT&SSFT (Maini et al.), Self-Filtering (Wei et al., 2022b), SELFIE (Song et al., 2019a), and RoCL (Zhou et al., 2021) exemplify this approach. However, these studies primarily focus on the dynamic changes of individual examples for the purpose of sample selection, rather than assessing the overall dynamics to distinguish different stages of the training process in learning with noisy labels.

---

[1] *"Memorization"* is not formally defined, and we denote *Memorization* as training DNNs to fit the assigned label for each particular instance, in a similar spirit as Feldman & Zhang (2020) and Forouzesh et al. (2023).

## 3    Determining Early Stopping by Tracking Fitting Performance

In this section, we aim to track the fitting performance to pinpoint the turning points of a model's generalization performance during the training process in the presence of noisy labels. To provide a clearer view of the model's fitting performance, we incorporate two metrics: stability and variability metrics. With these metrics, we identify and describe the key turning points of fitting performance, and elaborate on their correlation with generalization performance. We further introduce a transitional stage termed "*learning confusing patterns*" between the two turning points in fitting performance. Building on this foundation, we formally introduce our proposed *Label Wave* method, which determines an early stopping point by tracking the changes in the model's fitting performance. All experiments presented in this section use a standard ResNet-18 backbone. For detailed information regarding the experiment settings, please refer to Appendix B.

Prior to an in-depth discussion, we first clarify three basic concepts (Goodfellow et al., 2016): (i) *Test error*, an empirical measure of "generalization performance", indicates a model's ability to make accurate predictions on previously unseen data. (ii) *Training error*, an empirical measure of "fitting performance", indicates a model's ability to fit the training data. (iii) *Training process*: Given a model $f(\cdot; \theta)$ that is being trained on a noisy training set $\mathcal{D}$ comprised of $n$ labeled examples $(\boldsymbol{x}_i, y_i)$ where $i = 1, 2, ..., n$, the model's predicted label for each training instance $\boldsymbol{x}_i$ evolves as training progresses. After $t$ epochs, this predicted label for $\boldsymbol{x}_i$ is represented as $\hat{y}_i^t$.

### 3.1    Metrics for Tracking Fitting Performance

In Figure 2, we track both the *test error* and the *training error*. From our analysis of the *test error*, we identify two pivotal points where there is a notable shift in generalization performance: Point 1, where the test error attains its global minimum, and Point 2, where the test error transitions to a stable state after rising from this minimum. Notably, Point 1 is typically considered the optimal early stopping point. As determined by the *test error*, Point 1 indicates the model that generalizes best throughout the training process. Furthermore, an examination of the *training error* reveals that, after Point 1, the model starts to significantly fit mislabeled training examples. This suggests that fitting mislabeled examples impairs the model's generalization performance.

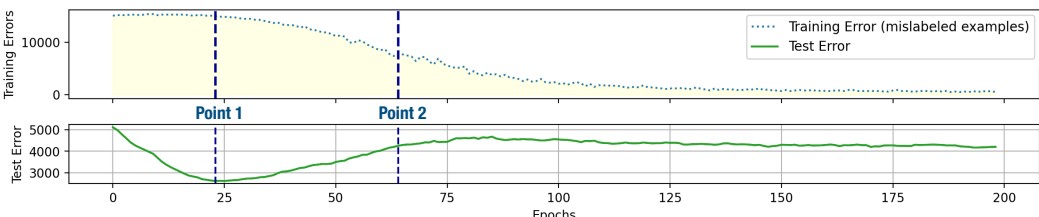

Figure 2: Tracking *test error* and *training error* (mislabeled examples) in training process.

In real-world applications, however, obtaining prior knowledge about which training examples are mislabeled examples is often impossible, thereby eliminating the possibility of relying on how well the model fits mislabeled examples as a criterion for early stopping. Furthermore, as we aim to eliminate the dependency on hold-out datasets, implementing early stopping based on validation errors becomes unfeasible. To address these challenges, we focus on the following two metrics to evaluate the model's "fluctuations in predictions" on the training set.

**Stability Metric.** We employ the *k-epoch learning* metric (Yuan et al., 2023) to quantify the stability of the model's predictions for specific subsets of examples, such as the clean data subset $\mathcal{D}_c$. A higher *k-epoch learning* value indicates that the model's predictions have greater stability in recent epochs and exhibit fewer fluctuations in predictions. Let $\text{acc}_i^t = \mathbb{1}_{\hat{y}_i^t = y_i}$ be a binary variable that indicates whether the model accurately classifies example $i$ at epoch $t$. The stability of the model's predictions from epoch $t - k + 1$ to epoch $t$ can be quantified as follows:

$$k\text{-}epoch\ learning = \sum\nolimits_{i \in \mathcal{D}_c} \text{acc}_i^t \wedge \text{acc}_i^{(t-1)} \wedge ... \wedge \text{acc}_i^{(t-k+2)} \wedge \text{acc}_i^{(t-k+1)}. \tag{1}$$

**Variability Metric.** In contrast to the stability metric, we use *prediction changes* as a variability metric to emphasize the model's inconsistency in classifying training set examples between the

current epoch and the previous one. A lower value of *prediction changes* indicates greater stability and fewer fluctuations in the model's predictions for the current epoch. The variability in the model's predictions from epoch $t - 1$ to epoch $t$ can be quantified as follows:

$$prediction\ changes = \sum\nolimits_{i \in \mathcal{D}} \mathbb{1}_{\hat{y}_i^t \neq \hat{y}_i^{(t-1)}}. \tag{2}$$

## 3.2 FITTING MISLABELED EXAMPLES IMPAIRS THE MODEL'S FITTING PERFORMANCE

Figure 2 presents two approaches to selecting the best generalizing model using both *test (validation) error* and *training error* on the mislabeled examples. However, these methods either require a separate hold-out set or rely on true labels in the training set. In contrast, our method focuses on a unique concept: *fluctuations in predictions* (FIPs). By quantifying these fluctuations using the stability and variability metrics, we can observe how the model's data-fitting strategy and its corresponding fitting performance change in two nearby points, which we defined in Section 3.1.

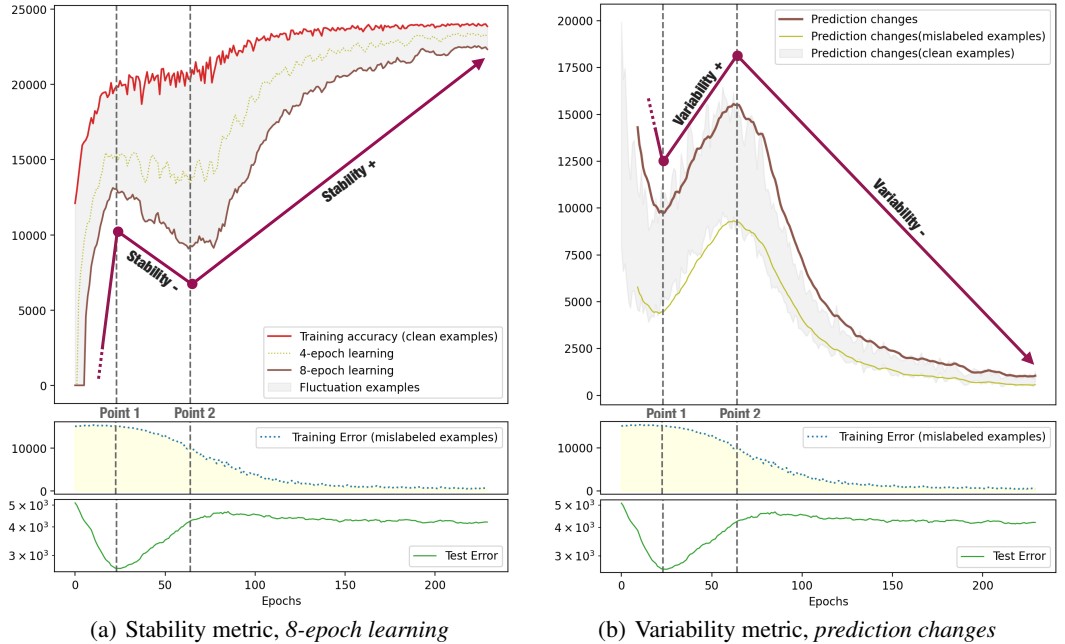

(a) Stability metric, *8-epoch learning*  (b) Variability metric, *prediction changes*

Figure 3: Using stability and variability metrics to track *fluctuations in predictions*.

In Figure 3, we chart these metrics throughout the training process of the model. Specifically, the stability metric is shown in Figure 3(a) and the variability metric in Figure 3(b). Both metrics are designed to track FIPs. Our observations highlight a clear correlation between the metrics' turning points and the critical points determined by the shifts in generalization performance. This allows us to establish some key empirical findings:

- Before Point 1, FIPs gradually decrease.
- After Point 1, FIPs shift from decreasing to increasing.
- After Point 2, FIPs shift from increasing to decreasing and eventually converge to a steady state.

Our empirical observation reveals a correlation between generalization performance and changes in the model's fitting strategy. We observe noticeable shifts in both performance and strategy before and after Point 1. After Point 1, the performance of models from subsequent epochs declines noticeably in their ability to fit the examples from the training set. Moreover, the data from our 8-*epoch learning* indicates that more than half of the examples in $D_c$ (clean training examples) cannot be stably predicted after Point 1. This significant decline contradicts the assumption that fluctuations following Point 1 only stem from a small subset of challenging examples within $D_c$. Therefore, we term the phenomenon where the change in the model's fitting performance at the Point 1 as *fitting mislabeled examples impairs the overall model's fitting performance*. This implies that by tracking *fluctuations in predictions*, we can identify whether the model has started to overfit mislabeled training data just by observing changes in its fitting performance.

Interestingly, the model's behavior from Point 1 to Point 2 is noticeably different from its behavior before and after these points. The stage from Point 1 to Point 2 cannot simply be considered as either "incorporation of prior" (Bengio et al., 2009) or "overfitting the dataset". Instead, we propose to see this as a transitional stage in learning with noisy labels, situated between the widely accepted stages of "*learning simple patterns*" (Arpit et al., 2017) and "*memorization*" (Zhang et al., 2017). We have coined this transitional stage "*learning confusing patterns*". A more detailed discussion on implications of this stage and Point 2 will be explored in Appendix A.

### 3.3 Label Wave Method

Based on our observations, we identified a correlation between the model's fitting and generalization performance when learning with noisy labels. With this understanding, we formally introduce our proposed *Label Wave* method. This method utilizes the observed correlation between *fluctuations in predictions* and *test error*. We aim to pinpoint an appropriate early stopping moment by closely tracking the model's predictions on the noisy training set. This moment, referred to as Point 1 in Section 3.1, is where the model begins to significantly fit mislabeled examples. To keep our method simple, intuitive, and computationally efficient, we directly refer to the *variability* metric explored in section 3.1, which we term as *prediction changes* or *PC*.

**Algorithm Flow.** We use the *prediction changes*, represented as $PC_t$, to track the changes in the model's fitting performance at the epoch $t$. The first turning point of $PC_t$ is identified as the early stopping point for model selection. In practice, tracking $PC_t$ is often susceptible to uncertainties inherent in the training process. We enhance the robustness of our *Label Wave* method by utilizing the *Moving Averages* techniques. By averaging $PC_t$ over the recent $k$ epochs (see Appendix E for the sensitivity analysis of $k$), we derive a more stable version, denoted as $PC'_t$, as follows:

$$PC'_t = (PC_t + PC_{t-1} + ... + PC_{t-k+1})/k. \tag{3}$$

As depicted in Algorithm 1, we compute $PC_t$ using the predictions of $f(\cdot; \boldsymbol{\theta})$ on the noisy training set $D$. By applying the *Moving Averages* and *Patience*, we identify the first local minimum point of the $PC'_t$ and designate it as the early stopping point for the model selection, as follows:

$$Early\ Stopping\ Point = t_{\text{first-min}}, \text{ where } t_{\text{first-min}} \text{ is the first local minimum of } PC'_t. \tag{4}$$

The *Label Wave* method allows us to pinpoint the moment when the model's fitting performance stability starts to decline by observing *fluctuations in predictions* and computing *prediction changes*. Based on our discussion in Section 3.2, stopping the training process at such a moment can prevent the model from beginning to overfit the mislabeled training data, ensuring the selection of a model with close-to-optimal generalization performance. This robust and intuitive method not only does not rely on validation sets and prior knowledge of training labels but also enhances the generalization of the selected model. To solidify our understanding and validate the efficacy of the *Label Wave* method, we embark on a series of empirical experiments detailed in the upcoming section. For scenarios where the *Label Wave* method does not work, see the discussion in Appendix C.3.

---

**Algorithm 1** Label Wave

---

Let $\boldsymbol{\theta}_o$ be the initial parameters and $v$ be the local minimum of PC. Let $p$ be the "*Patience*", representing the number of times a worsening PC is observed before halting.
$\boldsymbol{\theta} \leftarrow \boldsymbol{\theta}_o, t \leftarrow 0, i \leftarrow 0, v \leftarrow \infty$
1: **while** $i < p$ **do**
2:     Update $\boldsymbol{\theta}$ by running the training for $n$ steps, and $t \leftarrow t + n$.
3:     $PC_t \leftarrow$ Compute *prediction changes* (PC) in step $t$.
4:     $PC'_t \leftarrow$ *Moving Averages* PC in recent $k$ steps.
5:     **if** $PC'_t < v$ **then**
6:        $v \leftarrow PC'_t$ ; $i \leftarrow 0, \boldsymbol{\theta}^* \leftarrow \boldsymbol{\theta}, t^* \leftarrow t$    // Models stored at every new local minimum.
7:     **else**
8:        $i \leftarrow i + 1$    // Counting *Patience* when $PC'_t$ is larger than local minimum.
9:     **end if**
10: **end while**=0
Best parameters are $\boldsymbol{\theta}^*$, and best number of training steps is $t^*$.

---

## 4 EXPERIMENTS

In this section, we empirically demonstrate the effectiveness of our *Label Wave* method. In Section 4.1, we conduct comprehensive experiments to validate the broad applicability of our method across different settings. Section 4.2 highlights the capability of the *Label Wave* method to enhance the generalization performance of existing learning with noisy labels methods. Furthermore, in Section 4.3, we examine the Kendall $\tau$ correlation between our method and test accuracy. This analysis is compared with empirical validation on hold-out data. The findings suggest that *Label Wave* amplifies the generalization performance of learning with noisy labels methods not only by leveraging more training data but also by improving the precision in identifying early stopping points.

### 4.1 EFFECTIVENESS OF LABEL WAVE

Building upon the *Label Wave* method experiments conducted using the ResNet-18 backbone, we performed additional tests. These experiments demonstrate the consistent efficacy of our proposed *Label Wave* method across a wide range of settings, including multiple datasets, diverse network architectures, a range of parameters, various optimizers, and different levels and types of label noise. As evident in Table 1, there is only a slight difference between the test accuracy of models selected by the *Label Wave* method and the global maximum test accuracy during the training process. This indicates that the Label Wave method consistently selects models, irrespective of different components and parameters, that are at or near the global maximum in test accuracy. More experimental results and details of experiment settings can be found in Appendix C.

**Noise.** We assessed the capability of *Label Wave* method to handle learning from various levels and types of label noise. These include 10% to 80% of the incorrect labels with the *Instance-dependent* noise (abbreviated as Ins.) (Xia et al., 2020) and *Symmetric* noise (abbreviated as Sym.) (Van Rooyen et al., 2015). Notably, with *CIFAR-N* (using its "Random 1/2/3" and "worst" setting in *CIFAR-10N* and "worst" setting in *CIFAR-100N*, characterized by approximately 40% real-world noise) (Wei et al., 2021), we evaluate effectiveness of *Label Wave* method on real-world noise.

**Architectures.** We verified the effectiveness of our method in learning with noisy labels over several commonly used deep learning models, including *ResNet* (He et al., 2016), *VGG* (Simonyan & Zisserman, 2014), *Inception-v3* (Szegedy et al., 2016), and *DenseNet* (Huang et al., 2017).

**Datasets.** Our evaluations also confirmed the effectiveness of *Label Wave* method across multi-datasets. These datasets comprise seven vision-oriented sets: *CIFAR-10*, *CIFAR-100* (Krizhevsky et al., 2009), *CIFAR-N* (Wei et al., 2021), *Clothing1M* (Xiao et al., 2015), *WebVision* (Li et al., 2017), *Food101* (Bossard et al., 2014), and *Tiny-ImageNet* (Le & Yang, 2015), along with a text-oriented dataset: *NEWS* (Kiryo et al., 2017; Yu et al., 2019). We tested our method on class imbalanced datasets *CID-CE* and with using class imbalanced method *CID-LDAM* (Cao et al., 2019).

**Parameters & Optimizers.** The robustness of the *Label Wave* method is validated by testing it with different learning rates, batch sizes, random seeds, and optimizers. By adjusting the batch sizes to 64, 128, 256, learning rates to 0.01, 0.005, 0.001, random seeds to 1, 2, 3, 4, 5, and employing different optimizers such as *SGD with momentum* (Robbins & Monro, 1951; Polyak, 1964), *RM-Sprop* (Tieleman et al., 2012), and *Adam* (Kingma & Ba, 2014), we demonstrate that the *Label Wave* method remains resilient to reasonable variations in both parameters and optimizers.

Table 1: Differences (mean±std) among the model selection methods. Lower is better.

| Noise (Sym.) | 20% | 30% | 40% | 50% | 60% | 70% | 80% |
|---|---|---|---|---|---|---|---|
| Global Maximum (%) | 84.08±0.10 | 81.73±0.37 | 80.45±0.68 | 75.48±0.46 | 68.57±0.70 | 57.57±0.98 | 37.46±0.92 |
| Label Wave (%) | 83.43±0.32 | 81.48±0.36 | 80.15±0.67 | 75.20±0.73 | 68.21±1.02 | 56.64±1.57 | 37.08±0.95 |
| Difference | **0.65%** | **0.25%** | **0.30%** | **0.28%** | **0.36%** | **0.93%** | **0.38%** |
| Noise (Ins.) | 20% | 30% | 40% | 50% | 60% | 70% | 80% |
| Global Maximum (%) | 84.28±0.45 | 82.83±0.65 | 77.69±0.71 | 64.74±0.78 | 46.80±1.52 | 29.48±1.02 | 20.59±0.69 |
| Label Wave (%) | 83.99±0.72 | 82.68±0.45 | 76.81±0.76 | 63.87±0.68 | 45.92±2.00 | 28.87±0.69 | 20.08±1.06 |
| Difference | **0.30%** | **0.15%** | **0.88%** | **0.87%** | **0.88%** | **0.61%** | **0.50%** |

| Noise & Datasets | Random 1 | Random 2 | Random 3 | Ins. 10% | Sym. 10% | CID-CE | CID-LDAM |
|---|---|---|---|---|---|---|---|
| Global Maximum (%) | 83.21±0.23 | 82.86±0.24 | 83.10±0.25 | 85.91±0.23 | 85.46±0.12 | 56.03±0.55 | 61.80±0.93 |
| Label Wave (%) | 83.03±0.35 | 82.53±0.23 | 82.92±0.50 | 85.56±0.72 | 84.92±0.06 | 55.31±0.86 | 61.07±0.88 |
| **Difference** | **0.18%** | **0.13%** | **0.18%** | **0.35%** | **0.54%** | **0.72%** | **0.73%** |
| **Datasets** | CIFAR-10 | CIFAR-100 | CIFAR-10N | CIFAR-100N | NEWS | Tiny-ImageNet | Adam |
| Global Maximum (%) | 79.99±0.57 | 49.92±0.81 | 76.23±0.17 | 47.48±0.49 | 42.58±0.79 | 34.88±0.15 | 79.12±0.57 |
| Label Wave (%) | 79.54±0.91 | 49.39±0.69 | 75.98±0.49 | 47.04±0.38 | 42.06±1.15 | 34.20±0.42 | 78.92±0.66 |
| **Difference** | **0.45%** | **0.52%** | **0.25%** | **0.45%** | **0.51%** | **0.68%** | **0.21%** |
| **Architectures** | ResNet-18 | ResNet-34 | ResNet-50 | ResNet-101 | VGG-16 | Inception-v3 | Dense-121 |
| Global Maximum (%) | 79.86±0.52 | 79.81±0.48 | 79.44±1.33 | 77.35±0.40 | 78.00±0.53 | 57.23±0.52 | 66.09±0.22 |
| Label Wave (%) | 79.36±0.77 | 79.50±0.42 | 78.94±1.21 | 76.99±0.75 | 77.77±0.43 | 56.80±0.73 | 65.87±0.38 |
| **Difference** | **0.50%** | **0.31%** | **0.49%** | **0.36%** | **0.23%** | **0.43%** | **0.22%** |
| **Parameters** | LR. 0.01 | LR. 0.005 | LR. 0.001 | BS. 64 | BS. 128 | BS. 256 | RMSprop |
| Global Maximum (%) | 80.08±0.62 | 76.98±0.79 | 72.55±0.57 | 80.23±0.57 | 80.11±1.02 | 76.88±0.66 | 76.79±0.61 |
| Label Wave (%) | 80.38±0.57 | 76.67±0.95 | 72.14±0.55 | 80.59±0.41 | 79.69±0.68 | 76.24±0.49 | 76.22±0.95 |
| **Difference** | **0.30%** | **0.31%** | **0.40%** | **0.36%** | **0.43%** | **0.64%** | **0.57%** |

## 4.2 ENHANCING EXISTING LEARNING WITH NOISY LABEL METHODS

To verify the effectiveness of our method in practical applications, we apply our proposed *Label Wave* method within a range of state-of-the-art learning with noisy labels methods. Among the methods we have explored are robust loss functions (Feng et al., 2020), robust regularization (Xia et al., 2021; Wei et al., 2022a), label noise correction (Liu et al., 2020; Cheng et al., 2021), sparsity over-parameterization (Liu et al., 2022a), and the baseline Cross-Entropy (Rubinstein, 1999).

As shown in Tables 2 and 3, we conducted evaluations using ResNet-18 on CIFAR-10 and ResNet-34 on CIFAR-100 (Krizhevsky et al., 2009; He et al., 2016), both tainted with 40% *Symmetric* noise (abbreviated as Sym.) (Van Rooyen et al., 2015). Our primary assessment metric was the test accuracy of the selected model, determined by the average and standard deviation over five runs. Notably, the *Label Wave* method consistently outperformed conventional validation methods employing 5% to 30% hold-out data. This indicates that the *Label Wave* method works as a more effective early stopping method than conventional hold-out data validation, thus augmenting the generalization performance of models trained by existing learning with noisy label methods. Results under more experimental settings can be found in the appendix D.

Table 2: Test accuracy (mean±std) of each method on CIFAR-10 (with 40% Sym. label noise).

| Method | Val. 5% | Val. 10% | Val. 20% | Val. 30% | Label Wave |
|---|---|---|---|---|---|
| CE (Rubinstein, 1999) | 80.29±0.66% | 79.93±0.52% | 78.94±0.52% | 77.38±0.59% | **81.61±0.44%** |
| Taylor-CE (Feng et al., 2020) | 83.85±0.61% | 83.28±0.34% | 82.33±0.43% | 81.66±0.12% | **85.06±0.30%** |
| ELR (Liu et al., 2020) | 90.23±0.47% | 88.99±0.41% | 88.43±0.38% | 87.24±0.41% | **90.45±0.52%** |
| CDR (Xia et al., 2021) | 86.45±0.36% | 85.98±0.43% | 85.60±0.50% | 83.71±0.23% | **87.69±0.10%** |
| CORES (Cheng et al., 2021) | 87.26±0.32% | 86.9±0.25% | 85.78±0.69% | 85.17±0.57% | **87.74±0.13%** |
| NLS (Wei et al., 2022a) | 82.62±0.59% | 81.89±0.18% | 81.12±0.53% | 79.23±0.32% | **83.45±0.19%** |
| SOP (Liu et al., 2022a) | 86.40±1.04% | 86.80±0.77% | 87.12±0.19% | 86.37±0.66% | **88.42±0.38%** |

Table 3: Test accuracy (mean±std) of each method on CIFAR-100 (with 40% Sym. label noise).

| Method | Val. 5% | Val. 10% | Val. 20% | Val. 30% | Label Wave |
|---|---|---|---|---|---|
| CE (Rubinstein, 1999) | 49.69±0.89% | 48.42±0.80% | 45.96±1.17% | 43.65±0.53% | **50.96±0.30%** |
| Taylor-CE (Feng et al., 2020) | 56.67±0.35% | 56.48±0.18% | 55.23±0.50% | 54.38±0.29% | **57.64±0.28%** |
| ELR (Liu et al., 2020) | 64.53±0.36% | 62.05±0.50% | 60.09±0.74% | 57.49±0.42% | **65.36±0.39%** |
| CDR (Xia et al., 2021) | 61.93±0.38% | 60.82±0.38% | 58.27±0.50% | 54.90±1.03% | **63.34±0.15%** |
| CORES (Cheng et al., 2021) | 44.32±0.82% | 44.48±0.35% | 43.32±0.57% | 41.28±0.47% | **45.03±0.38%** |
| NLS (Wei et al., 2022a) | 57.08±0.62% | 55.89±0.26% | 54.09±0.57% | 52.82±0.28% | **58.05±0.15%** |
| SOP (Liu et al., 2022a) | 67.27±0.45% | 66.28±0.61% | 64.46±0.48% | 63.02±0.48% | **68.53±0.30%** |

### 4.3 Correlation with Test Accuracy

In this subsection, we provide evidence that our proposed method exhibits superior performance compared to hold-out validation across a diverse range of validation set sizes and noise rates. We employ the same experimental settings as in Section 3.

**Hold-out Set Sizes.** We utilized the same subset of training data to calculate both *prediction changes* (PC) and hold-out validation. This implies that we only use a limited-size subset from the training data to compute PC. In the lower subfigure of Figure 4(a), the Kendall $\tau$ correlation (Kendall, 1938) is illustrated, showcasing the association between the ranking of model selection and actual test accuracy for both methods. In the upper subfigure of Figure 4(a), we focus on the test accuracy of models selected by both the *Label Wave* and hold-out validation methods. Our experiments emphasize that increasing the size of the hold-out set improves its precision in model selection. However, as the size of the hold-out set increases, the data available for training decreases, leading to a reduction in test accuracy for candidate models obtained during the training process. These findings suggest that our proposed *Label Wave* method can select models more precisely and yield better test accuracy, even when computed on a smaller set.

**Noise Rates.** We further compare the test accuracy of models selected by our proposed *Label Wave* method and the test accuracy of models selected by hold-out validation in various set sizes with varying noise rates. As shown in Figure 4(b), we demonstrate the gain in model test accuracy selected by the *Label Wave* method. This underscores the robustness and superior efficacy of the *Label Wave* method, especially in high levels of noise and large set sizes scenarios.

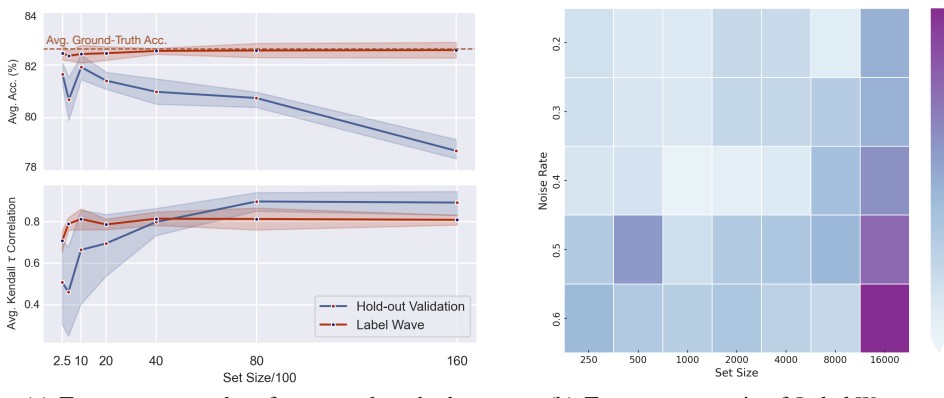

(a) Test accuracy and $\tau$ of compared methods     (b) Test accuracy gain of *Label Wave*

Figure 4: We aim to compare the test accuracy of models selected at the early stopping point by the *Label Wave* method and those selected by the hold-out validation method. (a) We selected a subset of the training data, with set sizes ranging from 250 to 16,000. This subset was used for both to compute the prediction change in the *Label Wave* method and serving as the hold-out set in hold-out validation, respectively. We computed the Kendall $\tau$ correlation and the test accuracy of the models selected by these two methods. (b) Further analysis was conducted to evaluate the difference in test accuracy between models selected by our proposed *Label Wave* method and those selected by the hold-out validation method with noise rates ranging from 20% to 60%.

## 5 Conclusion

In this paper, we introduced the *Label Wave* method for early stopping in the presence of label noise. Our proposed method uses the *prediction changes* metric to track changes in a model's fitting performance to pinpoint the early stopping point, eliminating the need for separate hold-out data. Extensive experiments showcased the effectiveness of the *Label Wave* method across various settings, leading to improved generalization performance in methods for learning with noisy labels. Furthermore, we introduced a transitional stage in learning with noisy labels, denoted as *learning confusing patterns*. Looking forward, there is potential to devise an advanced early stopping metric that could surpass the capabilities of the *prediction changes* metric, thereby enhancing the *Label Wave* method. A deeper dive into the transitional stage may offer pivotal insights into how deep neural networks behave when learning with noisy labels.

ACKNOWLEDGMENTS

The authors thank the anonymous reviewers and area chairs for their insightful and helpful comments. Suqin Yuan especially thanks Muyang Li, Runqi Lin, and Yingbin Bai who gave helpful advice during the rebuttal. Tongliang Liu is partially supported by the following Australian Research Council projects: FT220100318, DP220102121, LP220100527, LP220200949, and IC190100031.

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

# A    DISCUSSION ON *learning confusing patterns*

In this section, we delve into Point 2, where the model's test error transitions from increasing to stabilizing, by utilizing the metrics introduced in Section 3. Moreover, we introduce a novel transitional stage in learning with noisy labels, termed *learning confusing patterns*.

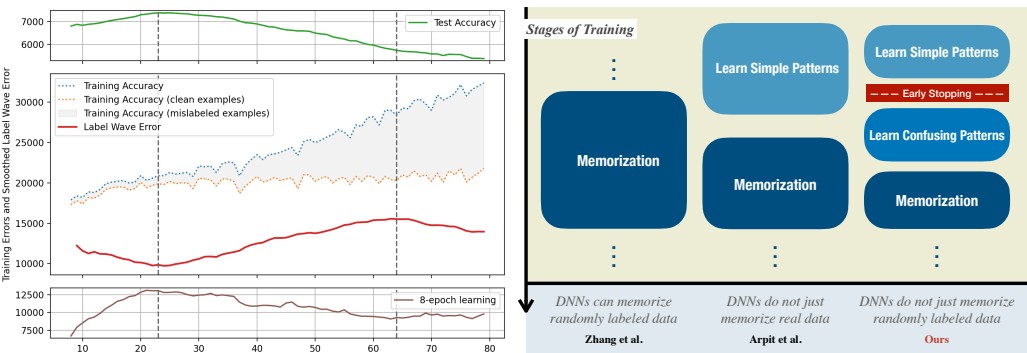

(a) Tracking model's behavior between two points      (b) Our proposed new transitional stage

Figure 5: Based on the multi-metrics we are tracking for the model's generalization and fitting performance between Point 1 and Point 2 (as shown in panel a), we propose a new transitional stage of learning with noisy labels, termed "*learning confusing patterns*" (shown in panel b).

As shown in Figure 5(a), when tracking *fluctuations in predictions* using stability (*8-epoch learning*) and variability (*prediction changes*) metrics, we noticed Point 2 is a turning point in the model's fitting performance, similar to the early stopping point, Point 1. Specifically, in Point 2, the *fluctuations in predictions* transitioned from increasing to decreasing trends, eventually stabilizing at a specific value. Here, we empirically explain the evolution of *fluctuations in predictions* during the training process. Before reaching Point 1, as training progressed, the model's generalization performance improved. The reduction in *fluctuations in predictions* suggests that a model with superior generalization is more noise-resistant (Arora et al., 2018; Morcos et al., 2018; Zhang et al., 2019; Forouzesh et al., 2023), resulting in more stable fits. After Point 2, the model began its shift towards memorization. This decrease in *fluctuations in predictions* reflects the beginning of the model fitting to the assigned label for each specific example (Arpit et al., 2017; Toneva et al., 2019; Feldman & Zhang, 2020; Xia et al., 2023), showing more consistent fits.

Nevertheless, the increase in *fluctuations in predictions* between Point 1 and Point 2 suggests the model is neither purely overfitting nor excellent in generalization during this stage. As shown in Figure 5(b), we describe this stage as the "*learning confusing patterns*", during which the stability of the model's predictions persistently decrease. Between these two points, the model's effort to minimize losses across the noisy training set causes it to adapt to mislabeled examples. This integration of incorrect information into generalization compromises the model's overall representation of the noisy training set, leading to an increase in *fluctuations in predictions* and a concurrent decline in both generalization and fitting performance. In contrasting the earlier stage before Point 1 where the model "learn from simple patterns", the learning paradigm during *learning confusing patterns* can be termed "learn from misled patterns", indicating that, during this stage, while the model still generalizes by learning patterns from the data, the patterns it learns are overly complex and incorrect due to the model try to generalize to mislabeled samples.

Building upon this, we identify Point 2 as the critical point where either generalization or memorization takes precedence. Beyond Point 2, the *fluctuations in predictions* begin to notably decline, indicating the model's shift towards fitting individual example labels and achieving stable fitting across the entire noisy training set. However, the marked transition at Point 2, from misguided generalization to overfitting, cannot be satisfactorily explained by solely relying on our predefined stability and variability metrics. This presents an intriguing open question for further investigation.

## B  DETAILED OVERVIEW OF BASIC MODEL IN OUR EXPERIMENTS

**Architecture**

- **Framework**: PyTorch, Version 1.11.0.
- **Model Type**: Standard ResNet-18, sourced from the PyTorch torchvision library.
- **Dropout**: Aligned with the standard ResNet-18, we do not incorporate dropout.

**Parameters**

- **Seed**: 1
- **Batch Size**: 128
- **Learning Rate**: Fixed at 0.01.
- **Optimizer**: Employs `optim.SGD` with momentum = 0.9.
- **Loss Function**: Utilizes the `CrossEntropyLoss` from the `nn` module.

**Dataset & Pre-processing**

- **Dataset**: The CIFAR-10 dataset, which is accessible via the `torchvision.datasets` module. 20% of the training data is held out for validation during the training process.
- **Normalization**: Leveraging the `torchvision.transforms` module, we normalize all pixel values to fit within the [0,1] range.
- **Cropping**: Images undergo random cropping procedures. For any 32x32 image, potential padding might be added, from which random 32x32 crops are then extracted.
- **Rotation**: Images are subjected to random rotations, limited to a range of ±15 degrees. This step ensures the model's robustness against various image orientations.

**Label Noise**

- **Symmetric Noise**: We infused 40% symmetric noise into 80% of the CIFAR-10 training set, affecting 40,000 examples.

## C   EFFECTIVENESS OF LABEL WAVE

In this appendix, we delve into the comprehensive details of experiments validating the effectiveness of the *Label Wave* method within various noisy label learning scenarios. We have presented specific results from each setting in the main content (see Table 1 in Section 4.1). However, here, we will discuss in-depth the experimental setups. Our experiments spanned a wide array of settings commonly employed in existing methods for learning with noisy labels. This encompassed varied architectures, datasets, noise types and levels, parameters, and optimizers. The overarching aim of these diverse setups was to gauge the effectiveness of the *Label Wave* method across different environments. Unless otherwise noted, the parameters and components remain consistent with the basic model in Appendix B outside of the each declared setting.

### C.1   NETWORK ARCHITECTURES

In our exploration, we employed a selection of prominent deep learning architectures to validate the versatility and effectiveness of the *Label Wave* method:

- **ResNet** (He et al., 2016): Renowned for its profound success in deep learning endeavors, especially in the realm of image classification, we engaged with multiple variants based on the original paper: *ResNet-18*, *ResNet-34*, *ResNet-50*, and *ResNet-101* .

- **VGG** (Simonyan & Zisserman, 2014): Known for its straightforward and effective architecture, we employed the *VGG-16*. The architecture's clarity and consistency enabled us to assess the *Label Wave* method's effectiveness with a well-established feature hierarchy.

- **Inception-v3** (Szegedy et al., 2016): With its unique design, *Inception-v3* offers diverse receptive fields without taxing computational resources. Using this architecture, we evaluated our *Label Wave* method in networks optimized for multi-scale feature extraction.

- **DenseNet** (Huang et al., 2017): By employing the densely connected convolutional architecture of *DenseNet-121*, we aimed to assess how the *Label Wave* method performs in a network characterized by robust gradient flow due to its dense connections.

### C.2   DATASETS EMPLOYED

To assess the *Label Wave* method's versatility, we employed several datasets, widely recognized in the learning with noisy labels domain:

- **Vision-oriented Datasets**:
  - *CIFAR-10* and *CIFAR-100* (Krizhevsky et al., 2009): Established benchmarks for image classification; CIFAR-10 contains 10 classes and CIFAR-100 contains 100 classes, respectively, and have 50,000 training examples and 10,000 test examples.
  - *CIFAR-N* (Wei et al., 2021): An augmented version of the CIFAR dataset, we use it "worst" setting, which is marked by approximately 40% real-world label noise, and presented an opportunity to test the resilience of the *Label Wave* method.
  - *Tiny-ImageNet* (Le & Yang, 2015): With 200 classes, each class contains 500 training images, making it a comprehensive dataset for evaluating performance.

- **Text-oriented Dataset**:
  - *NEWS* (Kiryo et al., 2017; Yu et al., 2019): We leveraged the NEWS dataset to examine the *Label Wave* method's applicability beyond images. The dataset consists of news articles categorized into various themes. For NEWS, we borrowed the pre-trained word embeddings from Pennington et al. (2014), and a 3-layer MLP is used with Softsign active function.

- **Real-world Scenarios Dataset**:
  - *CIFAR-N* (Wei et al., 2021): An augmented version of the CIFAR dataset, we use it "Random 1, 2, 3" setting, which is marked by approximately 20% real-world label noise, and presented an opportunity to test the resilience of the *Label Wave* method.

- *Clothing1M* (Xiao et al., 2015): A large-scale dataset that focuses on clothing classification. It contains over 1 million images of clothing items, categorized into 14 classes.
- *WebVision* (Li et al., 2017): Designed to mirror the real-world challenges of web-based image recognition, this dataset contains images collected from the internet. It comprises 2.4 million images from 1,000 different classes, mimicking the class distribution of ImageNet.
- *Food101* (Bossard et al., 2014): This dataset is focused on food recognition, consisting of 101 food categories, with 101,000 images. Each category contains 750 training images and 250 test images. The images are not artificially labelled and thus contain some level of real-world noise.

- **Class imbalanced dataset**: We tested our method on class imbalanced datasets, by setting the Imbalance Factor to 0.1 and the Noise Ratio (Sym.) to 0.4. Our experiments were conducted using our method in Cross-Entropy (Rubinstein, 1999) (*CID-CE*) and class imbalanced method LDAM (Cao et al., 2019) (*CID-LDAM*).

Table 4: Differences (mean±std) among the model selection methods. Lower is better.

| **Datasets** | *Clothing1M* | *WebVision* | *Food101* |
|---|---|---|---|
| Global Maximum (%) | 70.56±0.11 | 57.58±0.14 | 80.73±1.46 |
| Label Wave (%) | 70.12±0.34 | 57.24±0.34 | 80.12±1.01 |
| Difference | **0.44%** | **0.34%** | **0.61%** |

## C.3 LABEL NOISE

We examined the *Label Wave* method's effectiveness across various label noise conditions:

- **Symmetric Noise (Sym.)** (Van Rooyen et al., 2015): Symmetric noise levels were varied between 20% to 80%, which provided environments to examine how the *Label Wave* method works with data that has uniformly distributed noise across different labels.

- **Instance-dependent Noise (Ins.)** (Xia et al., 2020): By introducing 20% to 80% instance-dependent noise, we aimed to examine how *Label Wave* method works with data that has noise correlated with the instance characteristics.

- **Real-world Noise** (Wei et al., 2021): We have detailed the *CIFAR-N* dataset that is characterized by real-world noise. Building on that foundation, we have evaluated how *Label Wave* method performs when confronted with noise in real-world scenarios.

- **Low noise rate**: We augmented the CIFAR-10 dataset with 10% symmetric noise (*Sym. 10%*) and 10% instance-dependent noise (*Ins. 10%*), presenting an opportunity to test the resilience of the *Label Wave* method.

*Label Wave* **method is not applicable in very low or no label noise.** There are many situations where validation and test errors consistently decrease even with (low level) noisy labels in the training data. Modern deep neural networks often exhibit benign overfitting (Bartlett et al., 2020), a phenomenon also describable as a memorization effect (Zhang et al., 2017; Arpit et al., 2017).

The effectiveness of the Label Wave method in identifying an appropriate early stopping point is attributed to our design of a practical metric that tracks the significant onset of learning confusion patterns, namely prediction changes (PC). Therefore, if the training process lacks a stage of learning confusion patterns, such as when training with perfect data or employing robust regularization approaches, the original Label Wave method may not identify an appropriate stopping point.

However, it is important to note that in these scenarios, applying early stopping to improve the model's generalization performance might not be necessary.

## C.4 PARAMETERS AND OPTIMIZATION

Our experiments encompassed an array of parameters and optimizers:

- **Batch Sizes (BS.)**: *64, 128, 256*.
- **Learning Rates (LR.)**: *0.01, 0.05, 0.001*.
- **Optimizers**:
    - *SGD with momentum* (Robbins & Monro, 1951; Polyak, 1964): A widely-used optimizer, we gauged how its momentum-based optimization worked in tandem with the *Label Wave* method.
    - *RMSProp* (Tieleman et al., 2012): Known for its robustness and flexibility in non-stationary settings, we examined its synergy with the *Label Wave* method.
    - *Adam* (Kingma & Ba, 2014): Recognized for its adaptability in adjusting learning rates during training, we determine its synergy with the *Label Wave* method.
- **Seeds**: Each experiment is run five times, with the seed set to: *1, 2, 3, 4, 5*, respectively.

## C.5 REGULARIZATION TECHNIQUES

We examined the *Label Wave* method's effectiveness across various regularization techniques:

- **Mixup** (Zhang et al., 2018): Mixup is an innovative data augmentation technique that operates by creating virtual training examples. It linearly interpolates between pairs of examples and their labels, effectively encouraging the model to favor simple linear behavior in-between training examples.
- **Batch Normalization (BN)** (Ioffe & Szegedy, 2015): Batch Normalization is a widely adopted technique in deep learning, designed to stabilize and accelerate the training process. By normalizing the inputs of each layer across the mini-batch, it addresses the issue of internal covariate shift.
- **Dropout** (Srivastava et al., 2014): Dropout is a regularization technique that prevents overfitting by randomly deactivating a subset of neurons during training. By doing this, it effectively creates a network of varying architecture each time a batch is passed through, which discourages the network from relying too much on any single neuron and promotes better feature learning.
- **Data Augmentation (DA)**:
    - *Cropping*: Images undergo random cropping procedures. For any 32x32 image, potential padding might be added, from which random 32x32 crops are then extracted.
    - *Rotation*: Images are subjected to random rotations, limited to a range of ±15 degrees. This step ensures the model's robustness against various image orientations.

Table 5: Test accuracy (mean±std) of each techniques on CIFAR-10 (with 40% Sym. label noise).

| Regularization Techniques | Val. 20% | Label Wave | Global Maximum |
|---|---|---|---|
| Label Wave | 63.00±0.86% | 66.79±0.39% | 67.15±0.49% |
| Label Wave + BN + DA | 78.94±0.52% | 81.61±0.44% | 81.76±0.30% |
| Label Wave + BN + DA + Dropout | 81.30±1.07% | 83.57±0.24% | 83.77±0.32% |
| Label Wave + BN + DA + Mixup | 81.06±0.49% | 82.38±0.56% | 83.09±0.22% |
| Label Wave + BN + DA + Dropout + Mixup | 81.66±0.07% | 83.67±0.45% | 84.05±0.35% |

## C.6 QUANTITATIVE COMPARISON BETWEEN LABEL WAVE AND GLOBAL MAXIMUM TEST ACCURACY

In Table 1, we use "*Difference*" to indicate the disparity in test accuracy between the model that achieved the Global Maximum test accuracy throughout the training process and the model selected by the *Label Wave* method. This measure directly contrasts the performance of the *Label Wave* method with that of the ground-true best model (*Global Maximum*) across various settings. The *Difference* is determined by the following formula:

$$Difference = A_{GM} - A_{LW}$$

Where:

- $A_{GM}$ represents the average test accuracy of the model which have maximum test accuracy in the training process.

- $A_{LW}$ represents the average test accuracy of the model selected by the *Label Wave* method.

This formula provides a quantitative measure to determine how closely the model selected by the *Label Wave* method aligns with the optimal model (i.e., the model with Global Maximum test accuracy) for each setting. A smaller *Difference* indicates that the performance of the *Label Wave* method is near optimal, while a larger *Difference* suggests a divergence from the optimal performance.

Using the *Difference* metric, we objectively evaluate the performance of the *Label Wave* method across various settings. As shown in Table 1, the consistent and slight disparities between the models chosen by the *Label Wave* method and the optimal models (Global Maximum) underscore the effectiveness of the *Label Wave* method across diverse settings. This showcases the *Label Wave* method's aptitude in pinpointing an appropriate early stopping point for model selection, which aligns closely with or even matches the optimal models.

# D  ENHANCING EXISTING LEARNING WITH NOISY LABEL METHODS

As shown in Tables 6 and 7, we conducted evaluations using ResNet-18 on CIFAR-10 and ResNet-34 on CIFAR-100 (Krizhevsky et al., 2009; He et al., 2016), both trained with 40% *Instance-Dependent* noise (abbreviated as Ins.) (Xia et al., 2020). Further, as shown in Table 8, we added the different between the test accuracy of the model selected using the Label Wave method and the global maximum test accuracy during the training process when using various learning with noisy label methods with 40% *Symmetric* noise (abbreviated as Sym.) (Van Rooyen et al., 2015).

Table 6: Test accuracy (mean±std) of each method on CIFAR-10 (with 40% Ins. label noise).

| Method | *Val.* 10% | *Val.* 20% | *Label Wave* | *Global Maximum* |
|---|---|---|---|---|
| CE (Rubinstein, 1999) | 78.04±0.62% | 77.35±0.23% | 80.49±0.33% | 80.86±0.77% |
| Taylor-CE (Feng et al., 2020) | 81.30±0.14% | 81.09±0.43% | 83.24±0.49% | 83.63±0.60% |
| ELR (Liu et al., 2020) | 87.68±0.32% | 86.98±0.22% | 89.59±0.16% | 90.41±0.25% |
| CDR (Xia et al., 2021) | 85.66±0.43% | 84.86±0.25% | 87.42±0.51% | 87.63±0.22% |
| CORES (Cheng et al., 2021) | 79.45±0.38% | 78.77±0.35% | 81.22±0.64% | 81.64±0.65% |
| NLS (Wei et al., 2022a) | 80.61±0.80% | 80.25±0.25% | 82.36±0.86% | 82.63±1.24% |
| SOP (Liu et al., 2022a) | 84.08±0.25% | 83.51±0.30% | 85.57±0.35% | 85.92±0.15% |

Table 7: Test accuracy (mean±std) of each method on CIFAR-100 (with 40% Ins. label noise).

| Method | *Val.* 10% | *Val.* 20% | *Label Wave* | *Global Maximum* |
|---|---|---|---|---|
| CE (Rubinstein, 1999) | 42.59±0.21% | 41.82±0.64% | 44.86±0.62% | 45.23±0.11% |
| Taylor-CE (Feng et al., 2020) | 52.42±0.59% | 52.53±0.56% | 55.52±0.54% | 55.74±0.23% |
| ELR (Liu et al., 2020) | 64.90±0.33% | 64.61±0.03% | 66.85±0.11% | 67.33±0.57% |
| CDR (Xia et al., 2021) | 59.82±0.10% | 59.21±0.87% | 62.64±0.40% | 63.25±0.47% |
| CORES (Cheng et al., 2021) | 43.26±0.71% | 41.96±0.11% | 45.38±0.13% | 46.24±0.09% |
| NLS (Wei et al., 2022a) | 52.22±0.88% | 53.26±0.08% | 55.56±0.39% | 56.00±0.25% |
| SOP (Liu et al., 2022a) | 67.83±0.15% | 66.33±0.23% | 69.35±0.71% | 70.23±0.53% |

Table 8: Test accuracy (mean±std) of each method on CIFAR (with 40% Sym. label noise).

| CIFAR10 - Method | *Label Wave* | *Global Maximum* |
|---|---|---|
| CE (Rubinstein, 1999) | 81.61±0.44% | 81.76±0.30% |
| Taylor-CE (Feng et al., 2020) | 85.06±0.30% | 85.43±0.37% |
| ELR (Liu et al., 2020) | 90.45±0.52% | 90.76±0.70% |
| CDR (Xia et al., 2021) | 87.69±0.10% | 87.80±0.24% |
| CORES (Cheng et al., 2021) | 87.74±0.13% | 87.95±0.21% |
| NLS (Wei et al., 2022a) | 83.45±0.19% | 83.62±0.37% |
| SOP (Liu et al., 2022a) | 88.42±0.38% | 88.82±0.46% |

| CIFAR100 - Method | *Label Wave* | *Global Maximum* |
|---|---|---|
| CE (Rubinstein, 1999) | 50.96±0.30% | 51.05±0.33% |
| Taylor-CE (Feng et al., 2020) | 57.64±0.28% | 57.99±0.30% |
| ELR (Liu et al., 2020) | 65.36±0.39% | 66.33±0.93% |
| CDR (Xia et al., 2021) | 63.34±0.15% | 63.54±0.28% |
| CORES (Cheng et al., 2021) | 45.03±0.38% | 45.75±0.27% |
| NLS (Wei et al., 2022a) | 58.05±0.15% | 58.32±0.35% |
| SOP (Liu et al., 2022a) | 68.53±0.30% | 68.78±0.27% |

# E    SENSITIVITY ANALYSIS OF $k$ VALUE IN MOVING AVERAGES

In this appendix, we present a sensitivity analysis of the $k$ value used in computing moving averages within the framework of the Label Wave method. This method, as detailed in the paper, employs moving averages of Prediction Changes (PC) for early stopping in training models with noisy labels.

This analysis varied the $k$ value, which represents the number of epochs over which the moving average is calculated for the PC metric. We computed the Pearson Correlation Coefficient between the moving average of PC values (denoted as $\mathrm{PC}_t'$) and test accuracy for each $k$ value. The goal was to determine the impact of different $k$ settings on the relationship between the averaged PC values and the test accuracy. The table below summarizes the Pearson Correlation Coefficients for different $k$ values:

| $k$ Value | Pearson Correlation Coefficient |
|:---:|:---:|
| 1 | -0.8565 |
| 2 | -0.9435 |
| 3 | -0.9637 |
| 5 | -0.9367 |
| 10 | -0.9406 |

## E.1    OBSERVATIONS

- **Strong Negative Correlation:** A strong negative correlation was consistently observed across all $k$ values. This signifies a robust inverse relationship between the moving average of PC values and test accuracy.

- **Variation in Correlation Strength:** The correlation's strength varied with different $k$ values. The strongest negative correlation occurred at $k = 3$, while $k = 5$ and $k = 10$ exhibited a slight decrease in correlation strength.

## E.2    CONCLUSION

The sensitivity analysis reveals that the selection of $k$ value influences the strength of the correlation but does not significantly alter the overall negative relationship between the moving average of PC values and test accuracy.

