# OpenReview forum: "Early Stopping Against Label Noise Without Validation Data"
_ICLR.cc/2024/Conference — ICLR 2024 poster_

### Official Review · Reviewer_D1rG · 2023-10-17

**Soundness:** 3 good
**Presentation:** 3 good
**Contribution:** 3 good
**Rating:** 6
**Confidence:** 4

**Summary:**

Early stopping is one of the most prevalent approaches to select model. However, it requires additional validation data. This paper proposes a new method to early stop without using the validation data empirically.

**Strengths:**

- It does not require additional validation data for model selection.
- Experiments on various structures and settings.

**Weaknesses:**

- What authors proposed is only supported by the empirical result.
- Following the question 2, more discussions may be needed for the related previous researches.
- Since the metic is moveing average over k epochs, sensitivity analysis over k is needed.

**Questions:**

- It may not work well for extreme cases, e.g. class imbalanced dataset. Any solution for those settings? If not, some assumptions could be specified for the setting.
- What is the difference between the previouse studies'[1,2] findings and what authors propose in section 3.2 (fitting
mislabeled examples impairs the overall model’s fitting performance)?
- Will PC monotonically decrease before its local minima? In other words, are there no fluctuations? or should we need some threshold?
- How will this pattern change when utilized with additional regularizations e.g. data augmentation? Will it be consistent or will it flutuate before it goes to local minima?
- For Table 1 and Table 2, which algorithm is utilized (just Cross Entropy?)? If utilized with several algorithm managing noisy labels, how much different between best and label wave?
- Want to see result on more noise condition for table 3 and 4.
- How about on real noise, e.g. Clothing1M?
- Will this criterion fit to another task, e.g. semantic segmentation?

[1] Wei, J., Liu, H., Liu, T., Niu, G., Sugiyama, M., & Liu, Y. (2022, June). To Smooth or Not? When Label Smoothing Meets Noisy Labels. In International Conference on Machine Learning (pp. 23589-23614). PMLR.

[2] Cheng, H., Zhu, Z., Li, X., Gong, Y., Sun, X., & Liu, Y. (2020, October). Learning with Instance-Dependent Label Noise: A Sample Sieve Approach. In International Conference on Learning Representations.

---

> ### Author Response · Authors · 2023-11-18
> **Response to Reviewer D1rG (Part1)**
>
> We answer the questions one by one as follows. We are really open to further discussion.
>
>
> **Question.1 - Class imbalanced dataset**
> We have tested the performance of our method on class imbalanced datasets, by setting the Imbalance Factor to 0.1 and the Noise Ratio (Sym.) to 0.4, with other settings consistent with those described in Section 4.1. Our experiments were conducted using our method in Cross-Entropy (CE) and class imbalanced method LDAM [1].
>
> | Methods   | Label Wave | Global Maximum |
> |-----------|-----------------------|---------------------------|
> | CE        |      55.31±0.86%      |       56.03±0.55%         |
> | LDAM   |      61.07±0.88%      |      61.80±0.93%         |
>
> The experimental results affirm that our method can effectively identify early stopping points when learning from class imbalanced noisy datasets. While we emphasize the effectiveness of the Label Wave method in such scenarios, we acknowledge, as your intuition, that it may not perform equally well in more extreme cases. For example, when applied to the CIFAR-100 training set with an Imbalance Factor of 0.01 and a Sym. Noise Ratio of 0.4, the model's test accuracy exhibited significant fluctuations, leading to the ineffectiveness of the Label Wave method. We wish to clarify that our method is intended to identify appropriate early stopping points; thus, scenarios where early stopping itself is not applicable fall outside the scope of our research.
>
> [1] Learning Imbalanced Datasets with Label-Distribution-Aware Margin Loss, NeurIPS 2019.
>
> **Question.2 - Difference between the previous studies**
> Our research, along with the findings of [1, 2], aims to deepen the understanding of how a model's performance is impacted by fitting incorrectly labeled examples. However, we believe that these studies contribute to this understanding in different ways.
>
> Both [1] and [2] have introduced new methods to handle noisy labels based on their findings, and thus have made notable contributions to the field. Specifically, [1] discusses the effects of Label Smoothing (LS) and Negative Label Smoothing (NLS) in the context of label noise, mainly focusing on how to adjust label smoothing strategies. [2] tackles instance-dependent label noise, and concentrates on effectively sieving out wrongly labeled instances from the training data.
>
> While [1] and [2] have significantly contributed to the field by exploring how noisy labels affect a model's generalization performance; our research distinctly points out how fitting noisy labels can impact the model's fitting performance on the training set. In Section 3.2, we delve deeper into the specific effects of fitting mislabeled examples on the overall fitting performance on the training set. Our study, by designing metrics called ''stability'' and ''variability'', discusses how the model's fitting performance on the training set is influenced by fitting incorrectly labeled examples as the training progresses. Moreover, we highlight a transitional phase, which we term ''learning confusion patterns''. We believe that this perspective has not been fully explored in previous studies.
>
> [1] Wei, J., Liu, H., Liu, T., Niu, G., Sugiyama, M., & Liu, Y. (2022, June). To Smooth or Not? When Label Smoothing Meets Noisy Labels. In International Conference on Machine Learning.
>
> [2] Cheng, H., Zhu, Z., Li, X., Gong, Y., Sun, X., & Liu, Y. (2020, October). Learning with Instance-Dependent Label Noise: A Sample Sieve Approach. In International Conference on Learning Representations.

---

> ### Author Response · Authors · 2023-11-18
> **Response to Reviewer D1rG (Part2)**
>
> **Question 3 - PC Fluctuations & Sensitivity Analysis over k**
> Here, we address both Weakness 3 and Question 3. Based on your constructive suggestions, we have conducted additional analysis.
>
> 1. Monotonicity of PC Values:
> You inquired about the fluctuations in PC values and suggested setting some thresholds. We fully endorse your suggestion that applying a specific threshold to label wave in practice would further enhance the robustness of our method. In our existing experiments, we have found that Moving Averages and "Patience" are sufficient to accurately help us identity the appropriate early stopping point through PC. Therefore, to ensure the simplicity and straightforward of our proposed method, we did not add an additional threshold in this paper.
>
> 2. Sensitivity Analysis of k Values:
> In response to your suggestion, we conducted a sensitivity analysis on the k value for Moving Averages. Specifically, we analyzed the correlation between the PC after applying different k values of moving averages and test accuracy:
>
> | k   | Pearson Correlation Coefficient |
> |-----------|-----------------------|
> | 1        |      -0.8565      |
> | 2        |      -0.9435      |
> | 3        |      -0.9637      |
> | 5        |      -0.9367      |
> | 10        |      -0.9406     |
>
> (Experiment take the same settings as Sec. 4.1 and Appendix B)
>
> We observed that as the k value changes, the Pearson correlation coefficient between the moving average of PC values and test accuracy exhibits some fluctuations but overall maintains a very strong negative correlation. This suggests that while considering the setting of k values is necessary, different k values do not significantly alter the relationship between the moving average of PC values and test accuracy. We will provide detailed sensitivity analysis charts in the further revision.
>
>
>
> **Question.4 - Additional regularizations**
> Based on the reviewers' detailed comments, we tested the effectiveness of our Label Wave method with additional regularization methods. It is worth noting that our proposed Label Wave method itself includes BN and Data Augmentation, i.e., Label Wave + B, D as described below. Beyond the reviewer's comments, we have also tested the setting where these two regularization methods are not included. These regularization methods include:
>
> *A. Mixup
> B. BN
> C. Dropout
> D. Data Augmentation.*
> (All experiments take the same settings as Sec. 4.1)
>
> | Methods   | Label Wave | Global Maximum |
> |-----------|-----------------------|---------------------------|
> | Label Wave        |       66.79±0.39%      |    67.15±0.49%  |
> | Label Wave + B, D |  81.61±0.44%    |    81.76±0.30%    |
> | Label Wave + B, C, D     |   83.57±0.24%    |   83.77±0.32%    |
> | Label Wave + A, B, D      |   82.38±0.56%      |     83.09±0.22%      |
> | Label Wave + A, B, C, D  |83.67±0.45%|  84.05±0.35%     |
>
> The experimental results show that adding or subtracting these regularization methods in the experimental setting does not affect the effectiveness of the label wave method for identifying early stopping points.
>
>
> **Question.5 - LNL algorithms**
> Yes, Tables 1 and 2 evaluate the effectiveness of the label wave method when using Cross-Entropy Loss under various setting modifications.
> In Section 4.2, Tables 3 and 4 examine a range of algorithms for managing noisy labels. The differences between the maximum test accuracy and the label wave outcomes are shown below:
>
> | Methods   | CIFAR10 - Label Wave | CIFAR10 - Global Maximum |
> |-----------|-----------------------|---------------------------|
> | CE        |        81.61±0.44%               |81.76±0.30% |
> | Taylor-CE |        85.06±0.30%               |85.43±0.37%     |
> | ELR       |        90.45±0.52%               |90.76±0.70% |
> | CDR       |        87.69±0.10%               |87.80±0.24% |
> | CORES     |        87.74±0.13%               |87.95±0.21% |
> | NLS       |        83.45±0.19%               |83.62±0.37%     |
> | SOP       |        88.42±0.38%               |88.82±0.46%     |
>
>
> | Methods   | CIFAR100 - Label Wave | CIFAR100 - Global Maximum |
> |-----------|-----------------------|---------------------------|
> | CE        |        50.96±0.30%    |  51.05±0.33%   |
> | Taylor-CE |        57.64±0.28%    |  57.99±0.30%   |
> | ELR       |        65.36±0.39%    |  66.33±0.93%   |
> | CDR       |        63.34±0.15%    |  63.54±0.28%   |
> | CORES     |        45.03±0.38%    |  45.75±0.27%   |
> | NLS       |        58.05±0.15%    |  58.32±0.35%   |
> | SOP       |        68.53±0.30%    |  68.78±0.27%   |
>
>
> **Question.6 - More noise condition for table 3 and 4**
> Due to computational resource constraints, we will provide results with more noise condition for Tables 3 and 4 in subsequent Comments and further revision.

---

> ### Author Response · Authors · 2023-11-18
> **Response to Reviewer D1rG (Part3)**
>
> **Question.7 - Real noise**
> We respectfully wish to highlight that our testing on the CIFAR-10N [1] dataset involved human-annotated real-world noise, showcasing the Label Wave method's effectiveness in real-world scenarios. Here, we present the outcomes of applying our label wave method on the Clothing1M dataset.
>
> | Methods   | Label Wave | Global Maximum |
> |-----------|-----------------------|---------------------------|
> | CE        |      70.12±0.34%      |       70.56±0.11%         |
>
> [1] Wei, J., Zhu, Z., Cheng, H., Liu, T., Niu, G., & Liu, Y. (2021, October). Learning with noisy labels revisited: A study using real-world human annotations. In International Conference on Learning Representations.
>
>
> **Question.8 - Semantic segmentation**
> Considering that our aim is to demonstrate that the concept of "learning confusion patterns" helps detect the transitional point in learning with noisy labels, we did not consider tasks like semantic segmentation and focused solely on classification tasks.
>
>
> We extend our sincere thanks for your patience in reviewing our work and deeply appreciate your informative comments towards improving our manuscript.

---

> ### Comment · Reviewer_D1rG · 2023-11-20
>
> Thank authors for their sincere efforts to relieve my worries. I think this research has some interesing ideas, but I still have concerns as follows:
>
> - Since this strategy cannot be theoretically supported, many experimental results would be required.
> e.g. Which type of CIFAR10N the authors experimented? All 5 types? If so, results of all types should be reported. Although I agree the noisy label of CIFAR10N is realistic, since the image itself is 32$\times$32 CIFAR, it may be limited to represent more realistic situations.
> - I also like the result reporting type as Table 3 and 4. yet, results with more diverse settings of noise condition will be more convincing.
> - I think results with clean dataset will make this study more convincing.
>
> Therefore, currently I will keep my score as before.

---

> > ### Author Response · Authors · 2023-11-22
> > **Further reply to reviewer D1rG (Part 1)**
> >
> > We are encouraged that you think the paper has interesting ideas, and are glad to have relieved your worries. Here are our responses to your three extra concerns:
> >
> > **Q1:** Based on reviewers constructive feedback, further experimental results are shown below.
> >
> > | Settings   | Label Wave | Global Maximum |
> > |-----------|-----------------------|---------------------------|
> > | CIFAR-10N (Random 1)  |  83.03±0.35%    |    83.21±0.23%    |
> > | CIFAR-10N (Random 2)     |   82.53±0.23%    |   82.86±0.24%    |
> > | CIFAR-10N (Random 3)      |   82.92±0.50%      |     83.10±0.25%    |
> > | CIFAR-10 (Ins. 10%)  |85.56±0.72%|  85.91±0.23%     |
> > | CIFAR-10 (Sym. 10%)  |84.92±0.06%|  85.46±0.12%     |
> >
> > It is noteworthy that the regularization approach of our baseline model is sufficiently robust to overfitting on CIFAR-10N-aggregate. We observed an average test accuracy of 84.03±0.92% in 30 to 50 epochs, and 83.97±0.50% in 130 to 150 epochs, making early stopping unnecessary. Scenarios where early stopping is inapplicable fall outside the intended use of our method.
> >
> > We concur with your viewpoint that, although CIFAR-10N provides a fairly diverse and stable real-world testing environment, it may be limited in representing more realistic situations. We have also tested the applicability of our method on Tiny-ImageNet (image size 64x64) and clothing1M (image size >= 256x256), as shown in the table below.
> >
> > | Settings   | Label Wave | Global Maximum |
> > |-----------|-----------------------|---------------------------|
> > | Tiny-ImageNet |  34.20±0.42%    |    34.88±0.15%    |
> > | Clothing1M        |      70.12±0.34%      |       70.56±0.11%         |
> >
> >
> > **Q2:** We have supplemented the experiments in Tables 3 and 4 under the condition of instance-dependent label noise. The experimental results demonstrate that under the condition of instance-dependent label noise, our method can effectively enhance the performance of existing learning with noisy label methods.
> >
> > | Methods   | CIFAR10 - val. 10% | CIFAR10 - val. 20% |CIFAR10 - Label Wave | CIFAR10 - Global Maximum |
> > |-----------|-----------------------|-----------------------|-----------------------|---------------------------|
> > | CE       |      78.04±0.62%        |      77.35±0.23%          |   80.49±0.33%    |  80.86±0.77%   |
> > | Taylor-CE |     81.30±0.14%       |     81.09±0.43%       |  83.24±0.49%  |   83.63±0.60%  |
> > | ELR      |    87.68±0.32%        |     86.98±0.22%     |  89.59±0.16%   | 90.41±0.25%     |
> > | CDR       |    85.66±0.43%  |   84.86±0.25%     |  87.42±0.51%   |     87.63±0.22%    |
> > | CORES   |    79.45±0.38%       |    78.77±0.35%     |     81.22±0.64%       |  81.64±0.65%   |
> > | NLS       |    80.61±0.80%       |   80.25±0.25%      |     82.36±0.86%          |  82.63±1.24% |
> > | SOP       |      84.08±0.25%    |   83.51±0.30%      |     85.57±0.35%          |  85.92±0.15%   |
> >
> >
> > | Methods   | CIFAR100 - val. 10% | CIFAR100 - val. 20% |CIFAR100 - Label Wave | CIFAR100 - Global Maximum |
> > |-----------|-----------------------|-----------------------|-----------------------|---------------------------|
> > | CE       |       42.59±0.21%       |      41.82±0.64%         |      44.86±0.62%      |   45.23±0.11%  |
> > | Taylor-CE |       52.42±0.59%     |      52.53±0.56%      |       55.52±0.54%     |  55.74±0.23%   |
> > | ELR      |     64.90±0.33%       |   64.61±0.03%      |     66.85±0.11%       |   67.33±0.57%  |
> > | CDR       |     59.82±0.10%       |    59.21±0.87%    |    62.64±0.40%        |  63.25±0.47%   |
> > | CORES   |    43.26±0.71%       |    41.96±0.11%     |     45.38±0.13%       |   46.24±0.09%  |
> > | NLS       |       52.22±0.88%    |     53.26±0.08%    |   55.56±0.39%       |    56.00±0.25%        |
> > | SOP       |      67.83±0.15%     |    66.33±0.23%     |     69.35±0.71%          |   70.23±0.53%  |

---

> > ### Author Response · Authors · 2023-11-22
> > **Further reply to reviewer D1rG (Part 2)**
> >
> > **Q3:** We agree your viewpoint that if a method could be applied to identify early stopping points during the training of clean training data, it would further extend the applicability.
> >
> > However, as emphasized in our Contribution 1 (page 3 in paper), in this paper, we focus on introducing a new intermediate stage in learning with noisy labels, which involves learning confusion patterns. In this stage, model fitting mislabeled examples not only impairs the generalization performance but also the overall model’s fitting performance. In other words, the effectiveness of the Label Wave method in identifying an appropriate early stopping point is attributed to our design of a practical metric that tracks the significant onset of learning confusion patterns, namely prediction changes (PCs).
> >
> > Therefore, if the training process lacks a stage of learning confusion patterns, such as in the case of training perfect training data or when the regularization approach of the model is sufficiently robust, our method cannot locate a suitable early stopping point for the model. It is worth noting that often in these cases, early stopping itself is unnecessary, as modern deep neural networks often exhibit benign overfitting [1, 2].
> >
> > In scenarios where early stopping itself is unnecessary, if we still wish to use the Label Wave method to identify a stopping point for training, we can specify an additional threshold for the Label Wave method (as suggested in your Question 3). The training will stop when the total change in prediction changes (PCs) over t epochs becomes sufficiently small. In this regard, we have tested adding an additional threshold to the Label Wave method specifically for scenarios that do not involve learning confusion patterns. The training stops when the standard deviation between the prediction changes (PCs) over 20 consecutive epochs falls below 300 (CIFAR-10 with 50,000 training examples).
> >
> > | Settings   | Label Wave - threshold | Global Maximum |
> > |-----------|-----------------------|---------------------------|
> > | CIFAR-10-clean |  85.85±0.26%      |    86.48±0.13%    |
> > | CIFAR-10N-aggregate |      84.53±0.37%     |       85.37±0.12%      |
> >
> > We acknowledge that in scenarios where early stopping is unnecessary, pre-setting a stopping point or specifying a similar threshold for the change in training loss could achieve a similar effect. However, through our combined efforts with the reviewers, we have already tested numerous scenarios where early stopping significantly enhances the model's generalization performance. In these scenarios, using our Label Wave method can always bring additional benefits to the model training.
> >
> > [1] Benign overfitting in linear regression, PNAS 2020.
> > [2] Benign overfitting in classification: Provably counter label noise with larger models, ICLR 2023.
> >
> > In the end, we sincerely thank you for your further detailed constructive feedback, which enhance the practicality of our method in more scenarios.

---

> > ### Author Response · Authors · 2023-11-23
> > **Further reply to reviewer D1rG (Part 3)**
> >
> > To further address your concerns raised in extra Concern 1, we have tested our method on additional real-world datasets, including:
> >
> > 1. Food101, featuring images with a maximum side length of 512 pixels.
> > 2. WebVision, with images having a minimum side length of 256 pixels.
> >
> > For Food101, we employed the ResNet-50 model, and for WebVision, we employed the InceptionResNetV2 model. We kept the primary settings in line with those described in Section 4.1.
> >
> > Our findings are summarized in the table below:
> >
> > | Dataset   | Label Wave Accuracy | Global Maximum Accuracy |
> > |-----------|---------------------|-------------------------|
> > | Food101   | 80.12±1.01%         | 80.73±1.46%             |
> > | WebVision | 57.24±0.34%         | 57.58±0.14%             |
> >
> > These experiments validate the effectiveness of the Label Wave method on real-world datasets.
> >
> > Regarding the three extra Concerns you raised, we have already responded in our previous communications, "Further reply to reviewer D1rG (Part 1)" and "Further reply to reviewer D1rG (Part 2)". We hope these address your concerns.
> >
> > In the end, we sincerely thank you again for your constructive feedback and your efforts in enhancing our method.

---

> ### Comment · Reviewer_D1rG · 2023-11-23
>
> Thank the authors for their sincere efforts to address my concerns. With the inclusion of several additional experimental results, I am inclined to increase my score by +1 (as a result, 6). However, I would like to clarify that my acceptance of this paper is contingent upon the inclusion of the total experiments. As the current manuscript, without any modifications, falls below the acceptance threshold based on my previous scoring, I look forward to seeing the complete set of experiments in the revised version.

---

> > ### Author Response · Authors · 2023-11-23
> >
> > We sincerely thank you for your recognition of our work and the valuable suggestions you have provided. Your feedback is extremely important to us, and we have made corresponding modifications and additions to our Rebuttal Revision following your guidance.

---

### Official Review · Reviewer_R2TX · 2023-10-26

**Soundness:** 2 fair
**Presentation:** 3 good
**Contribution:** 3 good
**Rating:** 6
**Confidence:** 4

**Summary:**

The paper introduces an interesting method to perform early stopping in case of label noise without needing a validation set. The writing is good and the method provides some insights. However, there are concerns on the applicability of the method and the experiment set up is insufficient to evaluate the effectiveness of the method.

**Strengths:**

1. The method is interesting and has some insights.
2. The paper is well written, and everything is presented nicely.
3. Good results are shown on certain noise datasets.

**Weaknesses:**

1. My biggest concern is the applicability of the method. Currently it’s not clear the method works well on what scenarios.
2. Experiment evaluation is far from sufficient and the setup can be improved.

**Questions:**

1. Regarding applicability, my intuition is that the method only works well when there is “significant” amount of “random” label noise in the training set. “Random” is because the method relies on that the model fits simple patterns first and then learn random patterns from the noise. What if the label noise also only include simple patterns? E.g. black donkeys are mostly labeled as horses? The method also requires significant amount of label noise so that the model prediction fluctuate to a degree to be detected by the method. This is also reflected by the fact that experiments only considers >20% noise.  Can the authors provide more insights into this through discussion or experiments?
2. Only datasets with synthetic noise are considered. How does the method work on real datasets with real-world noise?
3. The baselines seem to use a noisy validation set, and evaluation is done on a clean test set. It makes more sense split the clean test set to create a clean validation set or to create a clean validation set from the training set. This is because we want to ensure the validation set has a same distribution as the test set, i.e. to be clean.
4. What happens if the amount of label noise is less than 20%? Including label noise from 0-20% can help better understand the method.
5. In the motivation of not using a validation set is that using a validation set reduces training set size and thus decreases performce, by this motivation, the method is targeting domains with limited amount of training set. How small the dataset size should be when it’s preferred to consider this method? Could you provide some analysis on this?
6. In order to show the method’s effectiveness, more datasets from diverse domains should be considered.

**Details Of Ethics Concerns:**

see questions

---

> ### Author Response · Authors · 2023-11-18
> **Response to Reviewer R2TX (Part1)**
>
> We would like to thank the reviewer for your detailed and insightful review. We are glad that you find the paper is interesting and has insights.
> Below we address the questions you’ve raised about the Label Wave method, and we remain open to further discussion if there are any additional comments on these questions.
>
> **Response to Question 1 and Question 4 (Applicability of the Label Wave method)**
>
> We appreciate the insightful concerns you have expressed about the applicability of the Label Wave method, particularly in scenarios with different types and amounts of label noise. We will detailedly address these concerns below in Part A - Types of Label Noise, and Part B - Amounts of Label Noise.
>
> *Part A: Types of Label Noise*
>
> 1. Broad Applicability Across Noise Types: Our method is effective not only in the presence of random label noise but also in more complex scenarios. We have conducted extensive experiments to prove our method's applicability to both symmetric noise and instance-dependent label noise [1], as well as in real-world environments with label noise [2, 3] (refer to Section 4.1, Tables 1 and 2). These experiments demonstrate that our method is capable of effectively identifying early stopping points across various types of label noise studied in the community of learning with noisy labels.
> 2. Limitations in Specific Noise Types: While we highlight the effectiveness of the Label Wave method, it is consistent with your intuition that our method may not work equally well across all types of label noise. Recent research [4] highlights a type of noise known as subclass-dominant label noise, where early stopping has proven to be ineffective. We wish to clarify that our method is designed to identify appropriate early stopping points; hence, scenarios where early stopping itself is inapplicable fall outside the scope of our research.
>
> *Part B: Amounts of Label Noise*
>
> 1. Effectiveness in Low Noise Scenarios: We would like to emphasize that our method remains effective in training sets with noise levels below 20%. Our experiments show that the method works well even with a noise rate as low as 10%. The choice to experiment with noise rates fixed between 20%-80% was based on the common range of dataset noise discussed in the research of learning with noisy labels.
> 2. Limitations at Very Low Noise Levels: However, we recognize that the concerns raised about the method's effectiveness at extremely low noise levels are valid. Our primary goal in this paper is to demonstrate how the concept of "learning confusion patterns" aids in detecting the transitional point in learning with noisy labels. To maintain simplicity, intuitiveness, and computational efficiency, we used the variability metric explored in Section 3.1, termed as prediction changes or PC (Section 3.3). Designing metrics that track the fluctuations of the model’s fitting performance on a broader range can be an effective way to improve our work. One initial intuition is to discuss the degree of fluctuations in prediction changes, which could help identify the early stopping point even when there is no obvious first local minimum in prediction changes.
> 3. 0% Label Noise: We have noted the suggestion to discuss scenarios where label noise is zero. However, as discussed in our paper, our method is not applicable to typical clean training sets. In cases of completely clean datasets, modern deep neural networks often exhibit benign overfitting [5, 6], making early stopping unnecessary. Scenarios where early stopping itself is inapplicable are beyond the scope of our method's intended use.
>
> [1] Part-dependent label noise: Towards instance-dependent label noise, NeurIPS 2020.
> [2] Learning with noisy labels revisited: A study using real-world human annotations, ICLR 2022.
> [3] Learning from Massive Noisy Labeled Data for Image Classification, CVPR 2015.
> [4] Subclass-Dominant Label Noise: A Counterexample for the Success of Early Stopping, NeurIPS 2023.
> [5] Benign overfitting in linear regression, PNAS 2020.
> [6] Benign overfitting in classification: Provably counter label noise with larger models, ICLR 2023.
>
>
> **Response to Question 2 and Question 6 (More datasets & Real-world noise)**
> We respectfully wish to highlight that our testing on the CIFAR-10N dataset involved human-annotated real-world noise, showcasing the Label Wave method's effectiveness in real-world scenarios. Furthermore, our experiments conducted on the NEWS and Tiny-ImageNet datasets have demonstrated our method's broad applicability across various domains. Here, we present the outcomes of applying our Label Wave method on the Clothing1M dataset. We are committed to supplementing more results using real-world datasets in the further revision.
>
> | Methods   | Label Wave | Global Maximum |
> |-----------|-----------------------|---------------------------|
> | CE        |      70.12±0.34%      |       70.56±0.11%         |

---

> ### Author Response · Authors · 2023-11-18
> **Response to Reviewer R2TX (Part2)**
>
> **Response to Question 3 (Validation set)**
> We really appreciate your wonderful suggestion about using a clean validation set to align with a clean test set distribution. However, in the context of our research on learning with noisy labels, we have intentionally chosen to maintain consistency between the distributions of the validation and training sets. This decision is grounded in the following considerations:
>
> 1. *Universality of Label Wave*: Our proposed method for identifying early stopping points using the Label Wave mthods aims to offer a universally applicable solution for learning with noisy labels. A prevalent assumption in this field, as reflected in numerous studies [7-10], is the presence of a training set with unreliable labels alongside a test set with reliable labels. In these cases, the training set is often divided into two parts: one for training and the other serving as a noisy validation set for early stopping. By adhering to this widely accepted setting, our method aligns with and can be directly compared with the method that utilizes a noisy validation set for identifying early stopping points.
> 2. *Label Wave in Real-World Applications*: In many real-world applications, it is common to encounter datasets where all available data is unreliable or noisy. This means that we cannot obtain an additional clean validation set for identifying early stopping points. Therefore, we only compare methods that utilize a noisy validation set for identifying early stopping points, as this reflects a more realistic and practical situation.
>
> In conclusion, we use a noisy validation set to ensure a fair and relevant evaluation of our method within the specific context of learning with noisy labels.
>
> [7] Early-learning regularization prevents memorization of noisy labels, NeurIPS 2020.
> [8] Robust Early-learning: Hindering the Memorization of Noisy Labels, ICLR 2021.
> [9] Mitigating Memorization of Noisy Labels by Clipping the Model Prediction, ICML 2023.
> [10] Late stopping: Avoiding confidently learning from mislabeled examples, ICCV2023.
>
>
> **Response to Question 5 (Training data size)**
> Although our method targets domains with limited training data, its fundamental principle of using all available data effectively applies universally when the training and validation sets have similar distributions. In such cases, incorporating the validation data into the training set can always be positive for the model performance, regardless of training data size or validation data split ratio.
>
>
> We really appreciate your feedback as it significantly contributes to refining our research. We are committed to investigating these aspects to extend the applicability of the Label Wave method.

---

> > ### Comment · Reviewer_R2TX · 2023-11-19
> > **Response to author**
> >
> > I thank the authors for clarifying the questions and also for the extra experiment on the Clothing 1M dataset. Most of my concerns are addressed but I have two extra questions:
> >
> > 1. The authors claims that 20%-80% label noise is what people commonly do, but not citations are provided. Also, some recent work also considers 0%-20% label noise[1]. Even if this is commonly done, it doesn't mean this is the correct way to do. For example, the CIFAR-10N dataset, using CIFAR-10N-aggregate or  CIFAR-10N-random (both have <20% label noise) is more realistic than CIFAR-10N-worse. I appreciate that the proposed method works well for heavy noise, but I do believe showing how it behaves in case of less noise is also important. I believe even if the method doesn't work well for less noise it doesn't undermine the paper's contribution, but instead the fact of presenting the results provides a more comprehensive view of the method.
> >
> > 2. The authors claim that for different training dataset size, incorporating the validation data into the training set can always be positive. When the training dataset size is very large, one can just use a fixed amount e.g. 10K data points for validation purpose. Since the training dataset size is large, including these extra 10k data makes little difference, but using the 10K data points can be very helpful for selecting the best epoch to prevent overfitting. I can imagine the proposed method is not needed if the dataset size is super large. The question here is that we want to figure out in what dataset size region, the method is useful and better than simply using a validation set. This is crucial for understanding the applicability of the method.
> >
> > [1] Wei, Jiaheng, et al. "Learning with Noisy Labels Revisited: A Study Using Real-World Human Annotations." International Conference on Learning Representations. 2022.

---

> > > ### Author Response · Authors · 2023-11-21
> > > **Further reply to reviewer R2TX**
> > >
> > > We are delighted to have addressed most of your concerns. Here are our responses to your two extra questions:
> > >
> > > **Q1:** Thank you for your suggestion regarding providing a more comprehensive view of the Label Wave method. Following your constructive feedback, to substantiate that the method works well even with a noise rate as low as 10%, we have now included experiments with <20% label noise, and the results are as follows:
> > >
> > > | Settings   | Label Wave | Global Maximum |
> > > |-----------|-----------------------|---------------------------|
> > > | CIFAR-10N (Random 1)  |  83.03±0.35%    |    83.21±0.23%    |
> > > | CIFAR-10N (Random 2)     |   82.53±0.23%    |   82.86±0.24%    |
> > > | CIFAR-10N (Random 3)      |   82.92±0.50%      |     83.10±0.25%    |
> > > | CIFAR-10 (Ins. 10%)  |85.56±0.72%|  85.91±0.23%     |
> > > | CIFAR-10 (Sym. 10%)  |84.92±0.06%|  85.46±0.12%     |
> > >
> > > It is noteworthy that the regularization approach of our baseline model is sufficiently robust to overfitting on CIFAR-10N-aggregate. We observed an average test accuracy of 84.03±0.92% in 30 to 50 epochs, and 83.97±0.50% in 130 to 150 epochs, making early stopping unnecessary. Scenarios where early stopping is inapplicable fall outside the intended use of our method.
> > >
> > > **Q2:** Thank you for clarifying the content of Question 5, which has prompted us to think more deeply about the applicability and limitations of our method. We concur with your intuition that when dealing with "sufficiently large" datasets, the benefits of Label Wave method in comparison to hold-out validation might not be as stark. However, the Label Wave method has no performance disadvantage in this case either.
> > >
> > > Figuring out the 'sweet spot' in terms of dataset size for Label Wave is tricky. It’s not just about how big the dataset is, but also what kind of data we're dealing with and how much of it is held out. Natural datasets often exhibit a long-tail distribution, and in such cases, maintaining or not the identical distribution between the validation set and the training set can affect the effectiveness of hold-out validation. Furthermore, as shown in Figure 4a of our paper, the proportion of hold-out data is also a key factor in discussing the effectiveness of the Label Wave method compared to hold-out validation. It is also challenging to answer your question through empirical validation, as we currently lack datasets that meet the "super large" condition for a more comprehensive empirical study.
> > >
> > > Although we are currently unable to precisely define the optimal size of datasets for which our method is best suited, we believe that this is an important research direction worth further exploration. It is noteworthy that our paper not only provides an effective early stopping method under conditions of limited training data but also introduces a new concept of "learning confusion patterns", which offers a new perspective for understanding and handling label noise.
> > >
> > > All in all, we sincerely thank your continued and positive feedback and thank your further constructive comments.

---

> > > > ### Comment · Reviewer_R2TX · 2023-11-21
> > > >
> > > > I thank the authors for answering my questions. The extra experiments and explanation address my concern. I will increase my score.

---

### Official Review · Reviewer_LTbz · 2023-10-31

**Soundness:** 3 good
**Presentation:** 3 good
**Contribution:** 2 fair
**Rating:** 5
**Confidence:** 4

**Summary:**

This paper studies an important topic of learning against noisy labels. Specifically, the authors mainly focus on how to automatical detect the transitioning point for early stopping, i.e., from fitting to clean to fitting to noise. There are two proposed key metrics so-called "stability" and "variability", and and the method uses "prediction change" as the mean of detecting early-stop point. The results show that the method can detect the point accurately by showing the test accuracy difference compared to that obtained from the global maximum point.

**Strengths:**

I felt there are multiple strengths of this work.
* n-depth analysis: This work provides phase 1 to phase 3 analysis to understand how the DNN learns knowledge from noisy data.
* Well-defined metric: Several useful metrics are proposed: prediction changes, stability, and variability.
* The paper is well organized and easy to read.

**Weaknesses:**

There are several weaknesses in this work, which may be useful to polish the paper.
* **Missing important reference:** I know references that are highly related to this work but unfortunately not mentioned and compared. These two papers also tackled exactly the same point and mentioned similar intuitions on what the best early stopping point is. These papers are worth mentioning and comparing.
[1] How does early stopping help generalization against label noise, arXiv 2019
[2] Robust learning by self-transition for handling noisy labels, KDD 2021

* **Unclear setup for practicality:** Detecting an early stop point is very important in the industry, especially when using strong regularization techniques together. For example, in the computer vision domain, using Mixup (or Cutmix, etc), Batch Norm, Dropout, and other architecture-specific regularization (e.g., stochastic depth for Vision Transformers) is a must-need. These kinds of strong regularization obviously change the learning behavior of DNNs like the training and testing curves. For the complete study toward practical methods, these recipes of training should be considered altogether, or theoretical support is needed.

These two major issues contribute the most when determining my review score.

**Questions:**

Please address the two major weaknesses.

**Details Of Ethics Concerns:**

No concern was detected.

---

> ### Author Response · Authors · 2023-11-18
> **Response to Reviewer LTbz**
>
> We extend our heartfelt gratitude to the reviewer for your detailed constructive feedback, addressing both the content of our paper and the applicability of our proposed method. Below we address the comments you’ve raised. We are very open to further discussion.
>
> **Weakness.1 - Related Work**
>
> In revised manuscript, we have added the references [1] and [2] in Related Work.
>
> [1] and [2] conceptualize model learning with noisy labels as two stages: the first stage learns from clean examples, and the second stage overfits to mislabeled examples. By identifying the largest safe set in the first stage and focusing on learning from that set in the latter stage, these methods effectively avoid the model's overfitting to mislabeled examples. Notably, [2] provides a method to identify the point where MP(t) equals MR(t) without additional supervision, thereby determining an early stopping point. [1] and [2] indeed provide good insights into the study of learning with noisy labels from two perspectives: the analysis of learning stages and the provision of practical learning with noisy label methods.
>
> Here, we will emphasize the advantages of our proposed method from two aspects: the learning stages analysis and the model selection method.
>
> *Learning Stages:*
>
> We introduce a novel stage between the stage dominated by learning simple patterns and overfitting mislabeled examples, named "Learning Confused Patterns". In this stage, the model begins to learn from mislabeled examples, leading to simultaneous declines in generalization and fitting performance.
>
> *Model Selection:*
>
> The Label Wave method not only requires no additional supervision but also has the advantages of being simple and efficient:
> 1. Simple Concept: It is based on the principle that the first local minimum in prediction changes marks the best early stopping point. Hence, the Label Wave method does not need to estimate the noise rate and is effective across a wide range of noise rates.
> 2. Direct Application: It tracks prediction changes during the training process, allowing the model to halt before learning confusing patterns. This approach avoids the complexity and computational costs of statistical models like the GMM and EM algorithms used in [2] and [3].
>
> [1] How does early stopping help generalization against label noise, arXiv 2019
> [2] Robust learning by self-transition for handling noisy labels, KDD 2021
> [3] Selc: self-ensemble label correction improves learning with noisy labels, IJCAI 2022
>
> **Weakness.2 - Strong Regularization Techniques**
>
> Based on the reviewers' detailed comments, we tested the effectiveness of our Label Wave method with additional regularization methods. It is worth noting that our proposed Label Wave method itself includes BN and Data Augmentation, i.e., Label Wave + B, D as described below. Beyond the reviewer's comments, we have also tested the setting where these two regularization methods are not included. These regularization methods include:
>
> *A. Mixup
> B. BN
> C. Dropout
> D. Data Augmentation.*
> (All experiments take the same settings as Sec. 4.1)
>
> | Methods   | Label Wave | Global Maximum |
> |-----------|-----------------------|---------------------------|
> | Label Wave        |       66.79±0.39%      |    67.15±0.49%  |
> | Label Wave + B, D |  81.61±0.44%    |    81.76±0.30%    |
> | Label Wave + B, C, D     |   83.57±0.24%    |   83.77±0.32%    |
> | Label Wave + A, B, D      |   82.38±0.56%      |     83.09±0.22%    |
> | Label Wave + A, B, C, D  |83.67±0.45%|  84.05±0.35%     |
>
>
> The experimental results show that adding or subtracting these regularization methods in the experimental setting does not affect the effectiveness of the label wave method for identifying early stopping points.
>
> In the end, we thank the reviewer for the two constructive comments mentioned above, and addressing them really improves our paper to a great extent.

---

> ### Comment · Reviewer_LTbz · 2023-11-20
> **Response to Authors**
>
> Thanks for the response from my questions. I carefully read the provided explanation and additional experiments. However, I still have some ambiguity.
>
> **Weakness.1 - Related Work**
>
> [1,2] will act as strong competitors to identify the early stopping point. I agree with that the proposed approach has distinct merits over them, but I am not sure if the proposed method outperforms the two existing works.
>
> **Weakness.2 - Strong Regularization Techniques**
>
> It was good to show effectiveness of the label wave even after applying data augmentation. But I wonder if the improvement still remains compared with Existing Label Noise Training without Label Wave. If strong augmentation are applied, the movement of test error will significantly change, e.g., the test error keeps going down over all training period. If this is the case, the early stopping is not required.
>
> I am also open to further discussion.

---

> > ### Author Response · Authors · 2023-11-21
> > **Further reply to reviewer LTbz**
> >
> > We appreciate the reviewers' timely follow-up and further constructive suggestions. Here are our responses to your two points of ambiguity:
> >
> > **Q1.** We appreciate your acknowledgment of the distinct merits of our Label Wave method compared to [1] and [2]. However, a direct comparison of its performance with methods [1, 2] is challenging. [1] require additional supervision either a small clean validation set or a noise rate τ, and the source code for method [2] is not publicly available.
> >
> > To ensure a fair comparison while maintaining comparability, we compare our proposed method with [1, 2] under the following settings:
> >
> > A. Following to the original setup of method [1], which involves using a small clean validation set for computation.
> >
> > B. Following the original setup of method [2], and under the assumption that the estimated noise rate τ is accurate.
> >
> > C. Modifying the setup of method [1], employing a small noise validation set for computation.
> >
> > D. Following the original setup of method [1], where an accurate noise rate τ is used to determine the early stopping point.
> >
> > | Methods   | Test Error (Lower is better) |
> > |-----------|-----------------------|
> > | Global Maximum |    21.59±0.06%           |
> > | Label Wave        |    21.73±0.13%         |
> > | A. [1] (clean val.) |    22.11±0.79%  |
> > | B. [2] |  23.10±0.17%       |
> > | C. [1] (noisy val.) |  23.26±0.73%  |
> > | D. [1] (noise rate) |     25.48±0.04% |
> >
> > (Other settings related to the experiment follow the open source code of [1]. Tested on CIFAR-10 using DenseNet-10-12.)
> >
> > Experimental results show that [1, 2] provide effective methods of identifying early stopping points. Compared with [1, 2], the Label Wave method can always pinpoint early stopping points more accurately.
> >
> > [1] How does early stopping help generalization against label noise, arXiv 2019
> > [2] Robust learning by self-transition for handling noisy labels, KDD 2021
> >
> > **Q2.** We address your concern about “Strong Regularization Techniques” below in Part A - improvement with hold-out validation, and Part B - test error significantly changed.
> >
> > *Part A - improvement with hold-out validation*
> >
> > Based on your constructive feedback, we also present the experimental results of using 20% of the available training data as hold-out validation to identify early stopping points, as shown in the following:
> >
> > *A. Mixup
> > B. BN
> > C. Dropout
> > D. Data Augmentation.*
> > (All experiments take the same settings as Sec. 4.1)
> >
> > | Methods   | 20% - Val  | Label Wave | Global Maximum |
> > |-----------|-----------------------|-----------------------|---------------------------|
> > | Label Wave        |         63.00±0.86%                |     66.79±0.39%      |    67.15±0.49%  |
> > | Label Wave + B, D |    78.94±0.52%   |81.61±0.44%    |    81.76±0.30%    |
> > | Label Wave + B, C, D  |           81.30±1.07%            |   83.57±0.24%    |   83.77±0.32%    |
> > | Label Wave + A, B, D  |     81.06±0.49%         |   82.38±0.56%      |     83.09±0.22%    |
> > | Label Wave + A, B, C, D  |          81.66±0.07%              |83.67±0.45%|  84.05±0.35%     |
> >
> >
> > *Part B - test error significantly changed*
> >
> > Yes, if the test error keeps going down over all training period, the early stopping is not required. Scenarios where early stopping itself is inapplicable are beyond the scope of our method's intended use.
> >
> > In conclusion, we sincerely thank you for your further constructive feedback, which enhance the practicality of our method.

---

> ### Comment · Reviewer_LTbz · 2023-11-21
> **Response.**
>
> Appreciate your hard working. The provided answer cleared my first concern, but still have concern for the second question. I saw many situations where validation and test errors are consistently going down even when there are noisy labels in training data (especially for real-world datasets, or when noise ratio is not significant).
>
> Assuming the test error is like Figure 1 sounds a bit strict. These test error movement trends may vary depending on the training setup (e.g. augmentation, optimizer, learning rate, architecture, etc.). I think it would be nice if the proposed method had more insights to add (or expand on) its advantages in all scenarios rather than simply mentioning something that is out of scope.
>
> I increased my score +1.

---

> > ### Author Response · Authors · 2023-11-22
> > **Further Response**
> >
> > Thank you for acknowledging our work and for your continued constructive feedback. We concur with your observations regarding the validation and test error trends in various training scenarios. Following your suggestions, we enhance the Label Wave method to identify suitable training stopping points in more general situations.
> >
> > 1.In practice, there are many situations where validation and test errors consistently decrease even with noisy labels in the training data, which indeed represents a more general case. We recognize that modern deep neural networks often exhibit benign overfitting [1, 2], a phenomenon also describable as a memory effect [3, 4].
> > In such cases, early stopping to enhance the model's generalization performance is unnecessary. However, even in these more general scenarios, a method to halt the training process is still needed.
> >
> > We noted your suggestion to add Label Wave advantages in all scenarios, which is indeed a very constructive recommendation. In response, we have incorporated a simple rule into the Label Wave method, enabling it to effectively determine an appropriate training stopping point even when validation or test errors are consistently declining.
> >
> > 2.As emphasized in our Contribution 1 (page 3 in paper), in this paper, we focus on introducing a new intermediate stage in learning with noisy labels, which involves learning confusion patterns. In this stage, model fitting mislabeled examples not only impairs the generalization performance but also the overall model’s fitting performance. In other words, the effectiveness of the Label Wave method in identifying an appropriate early stopping point is attributed to our design of a practical metric that tracks the significant onset of learning confusion patterns, namely prediction changes (PCs).
> >
> > Therefore, if the training process lacks a stage of learning confusion patterns, such as when training with perfect data or employing robust regularization approaches (as mentioned in your training setup, e.g., augmentation, optimizer, learning rate, architecture, etc.), the original Label Wave method may not identify an appropriate stopping point.
> >
> > 3.In the case your mentioned, the trend in prediction changes (PCs) mirrors that of validation or test errors, in other words, monotonic decrease. In such case, an appropriate training stopping point can be identified when the training's impact on model performance becomes “sufficiently small”. Thus, we have set an additional threshold for the Label Wave method. The training stops when the standard deviation between PCs over 20 consecutive epochs falls below 300 (CIFAR-10 with 50,000 training examples).
> >
> > | Settings             | Label Wave - threshold | Global Maximum         |
> > |----------------------|-----------------------|------------------------|
> > | CIFAR-10-clean       | 85.85±0.26%           | 86.48±0.13%            |
> > | CIFAR-10N-aggregate  | 84.53±0.37%           | 85.37±0.12%            |
> >
> > With these enhancements, the new Label Wave method effectively stops training even in the absence of a distinct learning confusion patterns stage. Note that the above enhancements do not affect the Label Wave method to find appropriate early stopping points in the presence of learning confusion patterns.
> >
> > 4.We acknowledge that in scenarios where early stopping is unnecessary, pre-setting a stopping point or specifying a similar threshold for the change in training loss could achieve a similar effect. However, through our combined efforts with the reviewers, we have already tested numerous scenarios where early stopping significantly enhances the model's generalization performance. In these scenarios, using our Label Wave method can always bring additional benefits to the model training.
> > Following your recommendations, we have refined the Label Wave method, making it a more robust tool for determining suitable training stopping points across diverse conditions.
> >
> > In the end, we sincerely thank you again for your thoughtful feedback and efforts towards improving our methods.
> >
> > [1] Benign overfitting in linear regression, PNAS 2020.
> >
> > [2] Benign overfitting in classification: Provably counter label noise with larger models, ICLR 2023.
> >
> > [3] Understanding deep learning requires rethinking generalization, ICLR 2017.
> >
> > [4] A closer look at memorization in deep networks, ICML 2017.

---

> > ### Author Response · Authors · 2023-11-23
> >
> > Dear reviewer LTbz,
> >
> > Sorry to disturb you. We would like to know are there any remaining issues that we did not solve, since we found that your current recommendation is weak rejection. Could you kindly let us know the point that makes you lean towards rejection. We will try our best to address your concerns in the remaining short discussion period and greatly appreciate your guidance to improve our work.

---

### Official Review · Reviewer_8gHt · 2023-10-31

**Soundness:** 3 good
**Presentation:** 3 good
**Contribution:** 2 fair
**Rating:** 6
**Confidence:** 4

**Summary:**

In the face of label noise, this publication presented an early halting technique using the Label Wave approach. Although the finding in this publication is intriguing, it only makes a little contribution to label noise bias.

**Strengths:**

This publication presented learning perplexing patterns, a transitional stage in learning with noisy labels.

**Weaknesses:**

The cifar10/100 dataset is utilized in this studies; however, real-world datasets such as webvision and food101 should be employed to confirm the efficacy of the suggested approach. I'm interested in seeing these outcomes.

In order to determine whether the suggested method chooses the best classifier, I would like to examine the maximum test accuracy during the training phase.

Given that the focus of this research is label noise, studies should compare the state-of-the-art techniques currently used for label noise learning, like DivideMix[1], ELR[2], AugDesc[3] and so on.

[1] Li, Junnan, Richard Socher, and Steven CH Hoi. "Dividemix: Learning with noisy labels as semi-supervised learning." arXiv
preprint arXiv:2002.07394 (2020).

[2] Liu, Sheng, et al. "Early-learning regularization prevents memorization of noisy labels." Advances in neural information
processing systems 33 (2020): 20331-20342.

[3] Nishi, Kento, et al. "Augmentation strategies for learning with noisy labels." Proceedings of the IEEE/CVF Conference on
Computer Vision and Pattern Recognition. 2021.

**Questions:**

See weaknesses.

---

> ### Author Response · Authors · 2023-11-18
> **Response to Reviewer 8gHt**
>
> We thank the reviewer for your comments. We are glad you point out that the ‘learning confusion patterns’ presented in our paper is intriguing. Below we address the concerns you’ve raised, and we’d be very open to further discussion.
>
> 1. **SOTA semi-supervised LNL methods on real-world datasets.**
>
> We would like to humbly point out that the CIFAR-10N dataset we have tested is a dataset labeled with human-annotated real-world noise. Besides, the experiments on NEWS and Tiny-ImageNet have demonstrated the applicability of our method on datasets from diverse domains. Meanwhile, our method is very simple and straightforward that can be used to strengthen existing methods, and thus cannot be directly compared with the state-of-the-art semi-supervised learning with noisy labels methods.
>
> To address the common concerns of the reviewers, and in order to improve the credibility of our method, we tested the efficacy of our method applied to semi-supervised methods on real-world datasets.
>
> Here, we present the outcomes of applying our Label Wave method in state-of-the-art semi-supervised methods, on the Clothing1M dataset. Notably, we have already evaluated ELR [2] in our paper (i.e., in Tables 3 and 4 on Page 8). Therefore, our current focus is to apply our Label Wave method to the methods including CE (baseline), DivideMix [1], and AugDesc [3]. We are committed to supplementing more results using real-world datasets and semi-supervised methods in the further revision.
>
>
> | Methods   | Label Wave | Global Maximum |
> |-----------|-----------------------|---------------------------|
> | CE        |      70.12±0.34%      |       70.56±0.11%         |
> | DivideMix |      71.71±0.50%      |       71.90±0.36%         |
> | AugDesc   |      73.91±0.60%      |       74.34±0.58%         |
>
>
> 2. **Maximum test accuracy during the training phase.**
>
> We would like to humbly point out that in Tables 1 and 2 (in Section 4.1), we presented a direct comparison between the test accuracy of models chosen through Label Wave and the maximum test accuracy model during the training phase. This comparison validates the effectiveness of our Label Wave method.
>
> For results in Tables 3 and 4 (Section 4.2), the difference between maximum test accuracy and label wave selected model are shown below:
>
> | Methods   | CIFAR10 - Label Wave | CIFAR10 - Global Maximum |
> |-----------|-----------------------|---------------------------|
> | CE        |        81.61±0.44%               |81.76±0.30% |
> | Taylor-CE |        85.06±0.30%               |85.43±0.37%     |
> | ELR       |        90.45±0.52%               |90.76±0.70% |
> | CDR       |        87.69±0.10%               |87.80±0.24% |
> | CORES     |        87.74±0.13%               |87.95±0.21% |
> | NLS       |        83.45±0.19%               |83.62±0.37%     |
> | SOP       |        88.42±0.38%               |88.82±0.46%     |
>
>
>
> | Methods   | CIFAR100 - Label Wave | CIFAR100 - Global Maximum |
> |-----------|-----------------------|---------------------------|
> | CE        |        50.96±0.30%    |  51.05±0.33%   |
> | Taylor-CE |        57.64±0.28%    |  57.99±0.30%   |
> | ELR       |        65.36±0.39%    |  66.33±0.93%   |
> | CDR       |        63.34±0.15%    |  63.54±0.28%   |
> | CORES     |        45.03±0.38%    |  45.75±0.27%   |
> | NLS       |        58.05±0.15%    |  58.32±0.35%   |
> | SOP       |        68.53±0.30%    |  68.78±0.27%   |
>
>
>
> In the end, we thank the reviewer for your detailed further revision comments that helped to improve the credibility of our method.

---

> > ### Author Response · Authors · 2023-11-23
> > **Response to Reviewer 8gHt (Part 2)**
> >
> > Following your advice, we have expanded the testing of our method to include a broader range of real-world datasets. These datasets are:
> >
> > *1. Tiny-ImageNet, with an image size of 64x64 pixels.*
> >
> > *2. Clothing1M, featuring images of at least 256x256 pixels.*
> >
> > *3. Food101, where the maximum side length of images is 512 pixels.*
> >
> > *4. WebVision, where the minimum side length of images is 256 pixels.*
> >
> > For Clothing1M and Food101, we employed the ResNet-50 model, while for WebVision, we employed the InceptionResNetV2 model. We maintained the primary settings consistent with those outlined in Section 4.1.
> >
> > The following table presents our findings:
> >
> > | Dataset       | Label Wave Accuracy | Global Maximum Accuracy |
> > |---------------|---------------------|-------------------------|
> > | Tiny-ImageNet | 34.20±0.42%         | 34.88±0.15%             |
> > | Clothing1M    | 70.12±0.34%         | 70.56±0.11%             |
> > | Food101   | 80.12±1.01%         | 80.73±1.46%             |
> > | WebVision     | 57.24±0.34%         | 57.58±0.14%             |
> >
> > These experiments further confirm the effectiveness of the Label Wave mehod on real-world datasets.
> >
> > In the end, we sincerely thank you again for your positive feedback and your efforts in enhancing our method.

---

### Author Response · Authors · 2023-11-18
**The Response to Reviewers**

Dear reviewers:

We sincerely appreciate all the reviewers for their thoughtful feedback and efforts towards improving our manuscript. We tried our best to address all mentioned concerns/problems. Are there unclear explanations here? We could further clarify them.

Best,
Authors

---

### Author Response · Authors · 2023-11-23
**The Response to All Reviewers**

Dear Reviewers, AC and SAC,

We would like to thank all of you again for taking your additional efforts to review our paper during this rebuttal phase, and for your valuable feedback.
We are heartened to know that our paper has been found interesting and insightful by the reviewers.
We have endeavored to thoroughly address all the concerns and issues raised, and we hope that our responses meet to your satisfaction.

Best regards,

The Authors

---

### Meta-Review · Area_Chair_9EHk · 2023-12-04

**Metareview:**

I have read all the materials of this paper including the manuscript, appendix, comments, and response. Based on collected information from all reviewers and my personal judgment, I can make the recommendation on this paper, *accept*. No objection from reviewers who participated in the internal discussion was raised against the accept recommendation.

**Research Question**

The authors address the noisy label problem with early stopping.

**Motivation**

Traditional methods rely on a validation set for model selection or early stopping, where the validation set is crucial and might lead to a sub-optimal selection due to its insufficient.

**Philosophy**

The authors aim to conduct early stopping without a validation set. To achieve this, the authors focused on the information derived through the training process.

**Technique**

The authors analyzed three stages of noisy label training and proposed a new measurement prediction change, which tracks the changes in the model’s predictions on the training set during the training  process, aiming to halt training before the model unduly fits mislabeled data. Although this paper does not have any theoretical analysis, the key idea is simple, neat, and elegant, which is also easy to use in practice.

**Experiments**

Extensive experiments demonstrate the effectiveness of the proposed method across a wide range of settings, including multiple datasets, diverse network architectures, a range of parameters, various optimizers, and different levels and types of label noise.

**Minor issues**

1. "Early stopping" or "early stop?" Please check.

2. The comma and period should be within the quotation mark, such as "early stopping."

3. For Figure 1, the bottom figure can be merged into the upper one. Although the authors provide long captions, the experimental setting is unclear. For example, which dataset, which loss function, what the noisy type and level?

4. Figure 2 is similar to Figure 1. The authors might consider to remove it.

5. I have a little confusion on stability metric. Based on my understanding, what I need is just the prediction change.

6. The figure along with Aloghtihm 1 can be removed.

7. It is better to show the prediction changes in the experimental part.

8. I have a same question with one reviewer, what the performance is on the clean data. Please add the extra results and discussions during the author-reviewer discussion period into the main paper or appendix.

9. The captions of Table 1 and 2 are the same. Please change them according to their exact contents.

**Justification For Why Not Higher Score:**

N/A

**Justification For Why Not Lower Score:**

This paper is simple, neat, and elegant. In my eyes it meets the bar of ICLR. The reviewer team did not have solid reasons to reject this paper.

---

### Decision · Program_Chairs · 2024-01-16

Accept (poster)